# CD137 (4-1BB) costimulation of CD8+ T cells is more potent when provided in cis than in trans with respect to CD3-TCR stimulation

Itziar Otano [1,2,3,13✉], Arantza Azpilikueta[1,3,4,13], Javier Glez-Vaz [1,3,4], Maite Alvarez [1,3,4], José Medina-Echeverz[5], Ivan Cortés-Domínguez[4,6], Carlos Ortiz-de-Solorzano [3,4,6], Peter Ellmark[7,8], Sara Fritzell[7], Gabriela Hernandez-Hoyos[9], Michelle Hase Nelson[9], María Carmen Ochoa [1,3,4,10], Elixabet Bolaños[3,4,10], Doina Cuculescu[1,3,4], Patricia Jaúregui[3,4], Sandra Sanchez-Gregorio[1,3,4,10], Iñaki Etxeberria [1,3,4], María E. Rodriguez-Ruiz[1,3,4,11], Miguel F. Sanmamed[1,3,4,12], Álvaro Teijeira[1,3,4,10,14], Pedro Berraondo [1,3,4,14] & Ignacio Melero [1,2,3,4,10,12,14✉]

CD137 (4-1BB; TNFSR9) is an activation-induced surface receptor that through costimulation effects provide antigen-primed T cells with augmented survival, proliferation and effector functions as well as metabolic advantages. These immunobiological mechanisms are being utilised for cancer immunotherapy with agonist CD137-binding and crosslinking-inducing agents that elicit CD137 intracellular signaling. In this study, side-by-side comparisons show that provision of CD137 costimulation in-cis with regard to the TCR-CD3-ligating cell is superior to that provided in-trans in terms of T cell activation, proliferation, survival, cytokine secretion and mitochondrial fitness in mouse and human. Cis ligation of CD137 relative to the TCR-CD3 complex results in more intense canonical and non-canonical NF-κB signaling and provides a more robust induction of cell cycle and DNA damage repair gene expression programs. Here we report that the superiority of cis versus trans CD137-costimulation is readily observed in vivo and is relevant for understanding the immunotherapeutic effects of CAR T cells and CD137 agonistic therapies currently undergoing clinical trials, which may provide costimulation either in cis or in trans.

[1] Program of Immunology and Immunotherapy, Cima Universidad de Navarra, Pamplona, Spain. [2] H12O-CNIO Lung Cancer Clinical Research Unit, Health Research Institute Hospital 12 de Octubre/ Spanish National Cancer Research Center (CNIO), Madrid, Spain. [3] Spanish Center for Biomedical Research Network in Oncology (CIBERONC), Madrid, Spain. [4] Navarra Institute for Health Research (IDISNA), Pamplona, Spain. [5] Bavarian Nordic GmbH, Planegg, Germany. [6] Program of Solid Tumours, Cima Universidad de Navarra, Pamplona, Spain. [7] Alligator Bioscience, Lund, Sweden. [8] Department of Immunotechnology, Lund University, Lund, Sweden. [9] Aptevo Therapeutics, Seattle, WA, USA. [10] Department of Immunology and Immunotherapy, Clínica Universidad de Navarra, Pamplona, Spain. [11] Department of Radiation Oncology, Clínica Universidad de Navarra, Pamplona, Spain. [12] Department of Oncology, Clínica Universidad de Navarra, Pamplona, Spain. [13] These authors contributed equally: Itziar Otano, Arantza Azpilikueta. [14] These authors jointly supervised this work: Álvaro Teijeira, Pedro Berraondo, Ignacio Melero. ✉email: iotanoan@alumni.unav.es; imelero@unav.es

CD137 (4-1BB) is a TNFR superfamily member (TNFRSF9) that was discovered as a costimulatory receptor acting on primed T cells[1] even though it is not exclusive to the T cell lineage but can be expressed by other leukocyte types[2,3]. Importantly, CD137 has a function in activated T cells that have been pre-exposed to TCR-CD3 (signal 1) and benefit from CD28 (signal 2) to gain intense CD137 surface expression[4]. In turn, when CD137 is ligated with its natural ligand (CD137L, 4-1BBL) or an agonist monoclonal antibody, it provides costimulatory signals for T cell proliferation, antiapoptosis, cytokine secretion, chromatin remodeling, and mitochondrial fitness[5]. The immunobiology of CD137-CD137L is mostly involved in antiviral cytotoxic T lymphocyte (CTL) responses as concluded from experiments in knock-out mouse strains[6,7] and from observations in patients deficient in CD137[8–10].

Costimulation signals via CD137 are mediated by TRAF2 and TRAF1 adaptors recruited to its cytoplasmic domain[11,12], which elicit a series of biochemical events primarily mediated by K63-polyubiquitination reactions which activate NF-κB and MAP kinase signaling pathways[12–14].

In mice bearing transplantable tumors, CD137 agonist monoclonal antibodies frequently induce tumor rejection and augment tumor-specific CTL responses[15,16] that are likely dependent on further crosslinking provided by Fcγ receptors. In human cancer patients, an antibody directed to CD137 termed urelumab, shows signs of clinical activity but induces serious liver inflammation in about one-fifth of treated patients at optimal doses[17,18]. To circumvent liver toxicity, targeting strategies to restrict CD137 exposure and crosslinking to the tumor microenvironment are already under clinical development as single agents or in combination with PD-L1 checkpoints inhibitors[5,19,20]. Such agents are mainly bispecific constructs targeting a molecule expressed by tumor cells, enabling CD137-costimulation in cis[19,21,22], or by stromal cells in the tumor microenvironment, such as fibroblast activated protein (FAP) enabling CD137-costimulation in trans[19]. PD-L1-CD137 bispecific antibodies are also under clinical and preclinical development[23] (GEN1046, ClinicalTrials.gov identifier NCT03917381).

Thus far, the most successful therapeutic use of CD137 signaling is observed in chimeric antigen receptors (CAR) encompassing the CD137 cytoplasmic domain[24] that in tandem with CD3ζ operating in-cis gives rise to critical signals for T cell function and persistence[25]. Previous evidence had shown that CD28 ligation in cis is far superior to that provided in trans with respect to TCR stimulation[26].

To assess the consequences of CD137-costimulation in cis or trans with respect to antigen-presentation, we experimentally compare the provision of CD137-costimulation in cis or in trans with respect to the source of CD3-TCR ligation. In this work, we show that cis-costimulation is more potent to invigorate T cell functions and is associated with more pronounced NF-κB signaling and transcriptomic changes favoring proliferation and DNA repair. As a consequence, the in vivo behavior of CD8+ T lymphocytes costimulated via CD137 provided in cis is qualitatively and quantitatively more efficient.

## Results

**CD137-cis and CD137-trans costimulation of human CD8+ T cells**. To mimic CD137-cis or CD137-trans costimulation, microbeads were covalently coated with either anti-CD3ε mAb, anti-CD137 mAb, or a mixture of both antibodies, leading to similar levels of mAb coating (Fig. 1a and b). These microbeads were used for a 96 hour-stimulation of CD8+ T cells isolated from the peripheral blood of multiple unrelated human donors, to study delivery of cis- versus trans costimulation. CD137-costimulation in cis resulted in a more prominent induction of

surface CD25 (Fig. 1c and e) and intracellular Bcl-xL expression (Fig. 1d and f).

Such superiority of cis-costimulation was preserved in a range of different densities of CD3 coating of the cis and trans beads and when using different bead to T cell ratios (Supplementary Fig. 1a–j). Of note, surface PD-1 expression was also more pronounced following cis-costimulation as compared to trans costimulation (Supplementary Fig. 2). Additionally, the secretion of IFNγ, IL-2 and granzyme B was superior in CD8+ T cells costimulated with microbeads providing CD137 ligation in cis as compared to costimulation in trans when cytokine secretion to the supernatants of these cultures was measured (Fig. 1g–i).

Conceivably T cell priming in vivo occurs first under the influence of CD28 costimulation. To model these conditions, CD8+ T cells were prestimulated with anti-CD3 plus anti-CD28 mAbs and rested without stimulation for 24 h prior to being exposed to the mAb coated microbeads. Superiority of cis-costimulation was also clearly observed in these CD28 prestimulated lymphocytes in terms of both activation markers and cytokine secretion (Supplementary Fig. 3).

The control of these lymphocyte functions has been related to the transcription factors T-bet and Eomes[27]. Accordingly, our experiments show that cis-costimulation results in a stronger induction of T-bet and reduction of Eomes expression resulting in a higher T-bet/Eomes ratio (Fig. 1j–l and Supplementary Fig. 1i). Recently, studies have reported on the critical metabolic control of CD137-costimulation on T cell mitochondrial mass and function[28–30]. As shown here, cis-costimulation rendered higher mitochondrial mass assessed by mitotracker staining (Fig. 1m) and more intense polarization of the mitochondrial membrane (Fig. 1n).

In addition to CD137 agonist antibodies as artificial ligands, we tested CD137L-Fc coated microbeads covalently built to compare cis- versus trans costimulation (Fig. 2a). Costimulation in cis by CD137L was superior to that provided in trans as measured for CD25 and Bcl-xL expression as well as secretion of IFNγ (Fig. 2b–e).

Considered together, these results indicate that all functional activities attributed to CD137 ligation on T cells are more potently induced when CD137 and CD3 stimulation are provided in cis by the microbeads experimentally resembling antigen-presenting cells.

Superiority of cis-costimulation could be explained by different access to the beads. To exclude such a possibility, we performed time-lapse microscopy imaging of fluorescence labeled CD8+ T cells in the presence of differently fluorescence stained cis and trans beads. Representative frames in Supplementary Fig. 4a and Supplementary movie 1, conclusively show that there is not steric hindrance for T cell to bead contact in the trans costimulation condition. Quantitative data supporting this notion are provided in Supplementary Fig. 4b. An additional analysis of the duration of individual transient beads to T cell contacts showed that contact times lasted equally in the different conditions (Supplementary Fig. 4c). In the same line, using beads coated with anti-CD7 mAb that binds to T cells without providing detectable costimulation did not impair costimulation given in cis by the CD3 CD137 beads (Supplementary Fig. 1f–j).

**Tumor cell-bound CD137-costimulation is superior in cis**. Bispecific monoclonal antibodies were used to study cis and trans CD137-costimulation as provided by tumor cells on which CD3-TCR and CD137 ligation could take place redirected by such bispecific protein constructs.

For this purpose, we used bispecific antibodies engaging CD3-EpCAM or CD137-5T4 (ALG.APV-527) and a colorectal carcinoma cell line HCT116 that coexpresses 5T4 and EpCAM on its surface. To model cis and trans CD137-costimulation with the

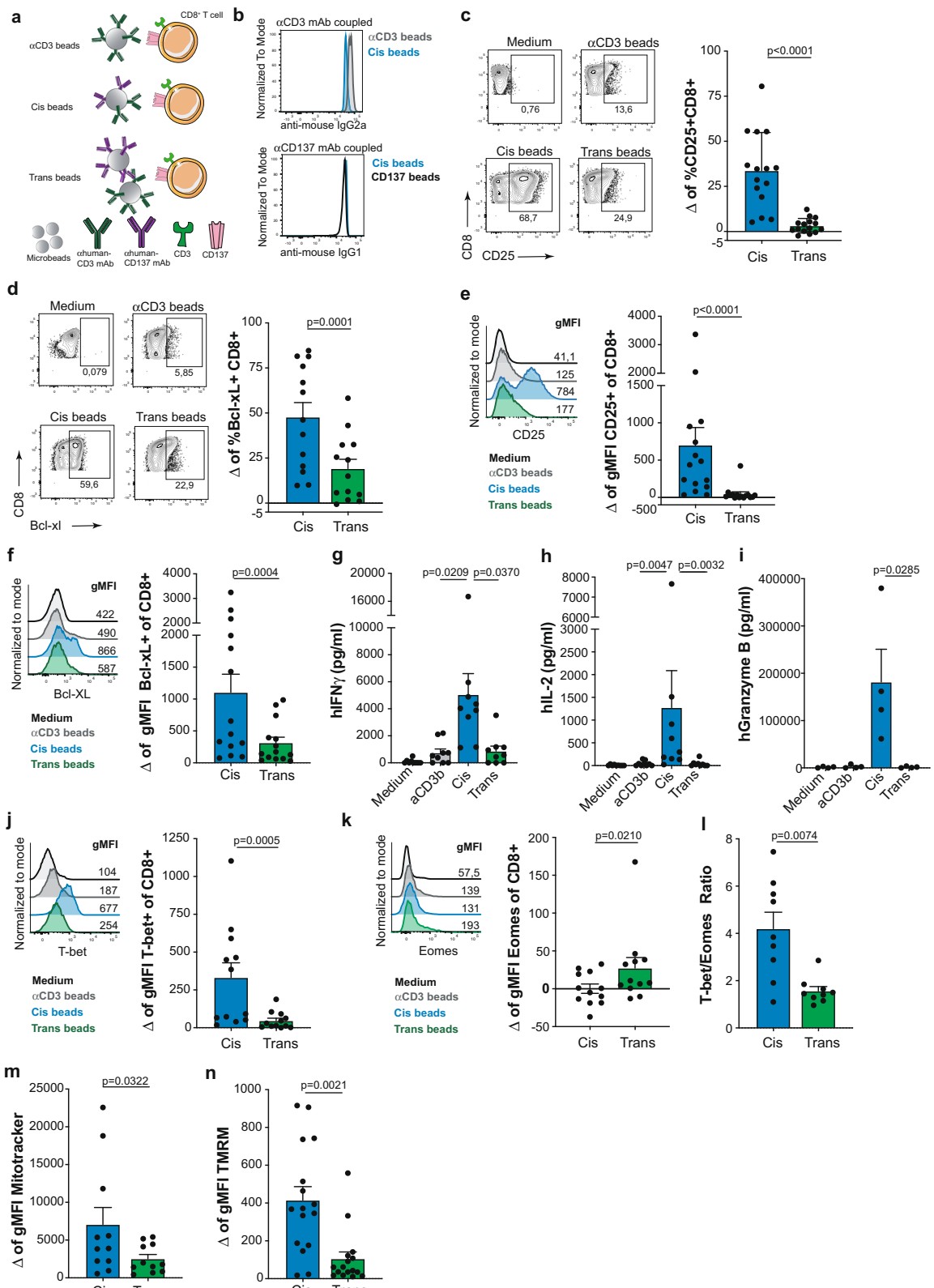

HCT116 cell line, the genes encoding either for EpCAM or 5T4 were silenced using CRISPR/Cas9 technology as shown in Fig. 3a and Supplementary Fig. 5a. Indeed, several clones of EpCAM and 5T4 silenced variants were obtained. Usage of such variants as compared to the wild-type (WT) cell line allowed to study cis-versus trans costimulation in the combined presence of CD3-EpCAM and ALG.APV-527 BsAb (Fig. 3b). In these co-culture

settings, cis CD8+ T cell costimulation using the WT HCT116 cell lines was compared to trans costimulation when plating 1:1 mixtures of the alternatively silenced variant clones (trans1 and trans2). In these settings, superiority of cis-costimulation was observed based on CD25 expression on CD8+ T cells and secretion of IFNγ, IL-2 and granzyme B (Fig. 3c–f). Differences in CD8+ T lymphocyte mitochondrial content were also observed

**Fig. 1 Comparative functional consequences of CD137-costimulation of human CD8+ T cells provided in cis versus trans. a** schematic representation of antibody-coated microbeads used to stimulate CD8+ T cells. **b** semiquantitative FACS assessment of anti-CD3 and anti-CD137 mAbs coupled into the microbeads was measured by antimouse IgG2a and antimouse IgG1 mAbs staining and detected by FACS. Representative histograms of the geometric mean fluorescence intensity for each condition. **c**, human primary CD8+ T cells from healthy donors were activated with mAb coated microbeads for 96 h. Representative dot plots of cell-surface CD25 (*n* = 15) and **d**, intracellular Bcl-xL (*n* = 13) expression analysed by flow cytometry in the indicated conditions. Summary data are given as delta (Δ), the difference of the value between each costimulation condition subtracted from the anti-CD3 condition in each case. **e** representative histograms of CD25 (*n* = 15) and **f**, Bcl-xL (*n* = 14) expression in CD8+ T cells. Numbers in the FACS histograms represent the geometric mean fluorescence intensity (gMFI). Summary data are given. Concentrations of human IFNγ (*n* = 9) (**g**) IL-2 (*n* = 8) (**h**) and granzyme B (*n* = 4) (**i**) in the culture supernatants of the indicated experimental conditions. **j**, representative histograms of T-bet and **k**, Eomes intracellular analyzed by flow cytometry at the indicated conditions (n = 12). Summary data are given as the difference of the value between each costimulation condition subtracted from the anti-CD3 value in each case. **l**, summary data showing the T-bet:Eomes ratio of CD8+ T cells in cis versus trans costimulation (*n* = 12). **m**, flow cytometry measurement of Mitotracker (*n* = 11) and **n** TMRM (Tetramethylrhodamine, Methyl Ester, Perchlorate) (*n* = 16) stainings given as geometric mean fluorescence in FACS-gated CD8+ T cells after cis versus trans costimulation. Data are given as mean ± s.e.m. Statistical significance was determined with the paired *t* test (two-sided) in **b**, **c** and **k**, Wilcoxon (two-sided) in **d**, **e**, **i**, **j**, **l** and **m**, the Friedman test (one-sided) with Dunn's correction in **f**, **g** and **h**. Source data are provided as Source Data file.

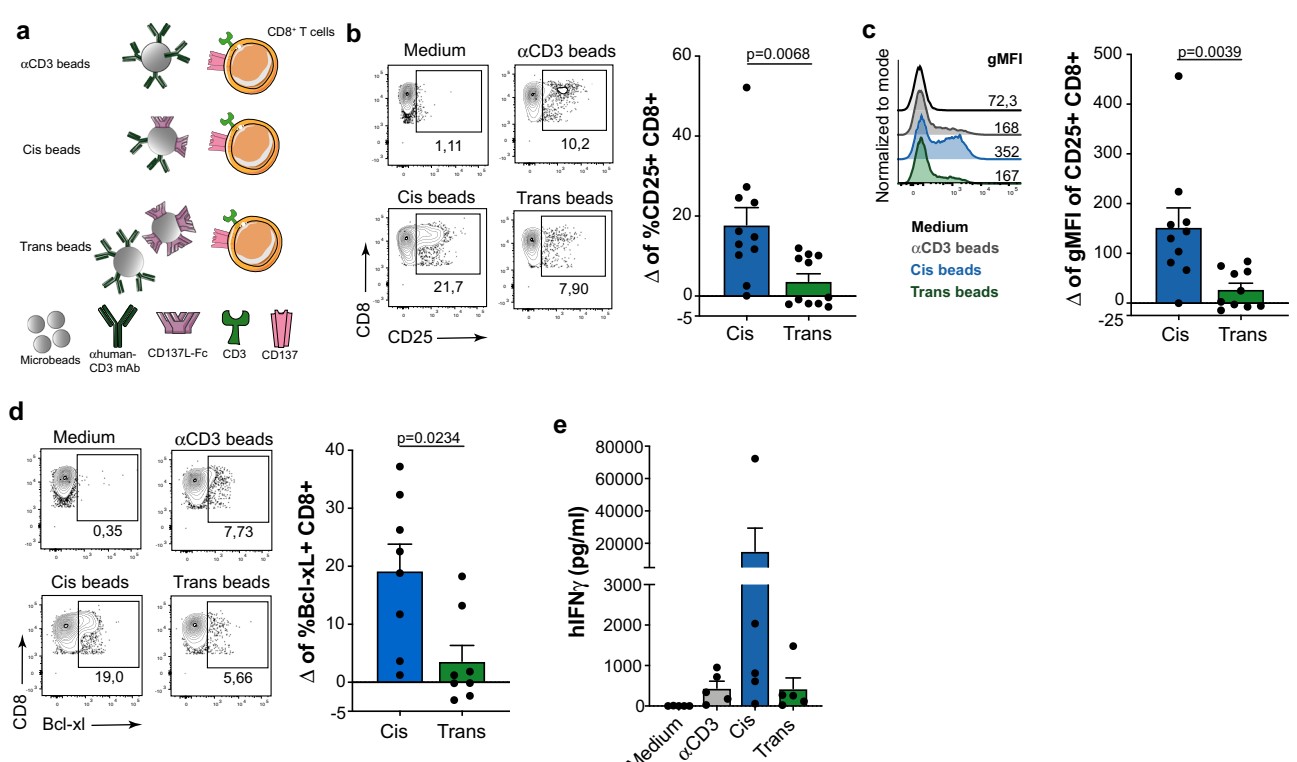

**Fig. 2 Cis versus trans CD137-costimulation by CD137L-Fc coated beads on human CD8+ T lymphocytes. a**, schematic representation of microbeads coated with CD137L-Fc used to stimulate CD8+ T cells. Human primary CD8 T cells from healthy donors were activated with microbeads for 96 h. **b**, representative dot plots of cell-surface CD25 (*n* = 11) and **c**, representative histograms (*n* = 10) of geometric mean fluorescence intensity (gMFI) for CD25 surface expression on CD8+ T cells. **d**, intracellular Bcl-xL (*n* = 8) expression analysed by flow cytometry in the indicated conditions. Summary data are given as delta (Δ), the difference of the value between cis and trans conditions to which the anti-CD3 background has been subtracted in each case. **e**, concentrations of human IFNγ (*n* = 5) in the culture supernatants of the indicated conditions. Data are given as mean ± s.e.m. Statistical significance was determined with the Wilcoxon test (two-sided). Source data are provided as Source Data file.

comparing cis- versus trans costimulation in the co-cultures clearly favouring the cis conditions (Fig. 3g).

Lack of T cell stimulation was observed in EpCAM silenced variants incubated with the CD3-EpCAM BsAb, confirming minimal levels of CD8+ T cell stimulation as measured by IFNγ and IL-2 production (Supplementary Fig. 5b and c).

Similarly to our observations with beads, interactions of CD8+ T lymphocytes with tumor cells providing cis or trans CD137-costimulation were comparable in frequency and cell to cell contact duration under live confocal microscopy analyses (Supplementary Movie 2 and Supplementary Fig. 6a–c). These experiments exclude less frequent intercellular interactions as a trivial reason for trans inferiority.

**CD137 cis-costimulation provides more intense NF-κB signaling.** To study in more depth the differences between cis versus trans CD137-costimulation, we tested NF-κB signaling upon incubation with the mAb coated microbeads and the alternatively silenced HCT116 cells for EpCAM and 5T4 in the presence of CD3-EpCAM and ALG.APV-527 BsAb.

A Jurkat cell line transfected in a stable fashion to constitutively express surface human CD137 and firefly luciferase under the control of an NF-κB sensitive promoter was co-cultured with the coated beads providing CD3 and CD137 stimulation either in cis or in trans. Independently of the source of stimulation, coated anti-CD137 mAb (Fig. 4a) or CD137L-Fc (Fig. 4b), the provision of CD137-costimulation in cis was far superior in terms of NF-κB dependent

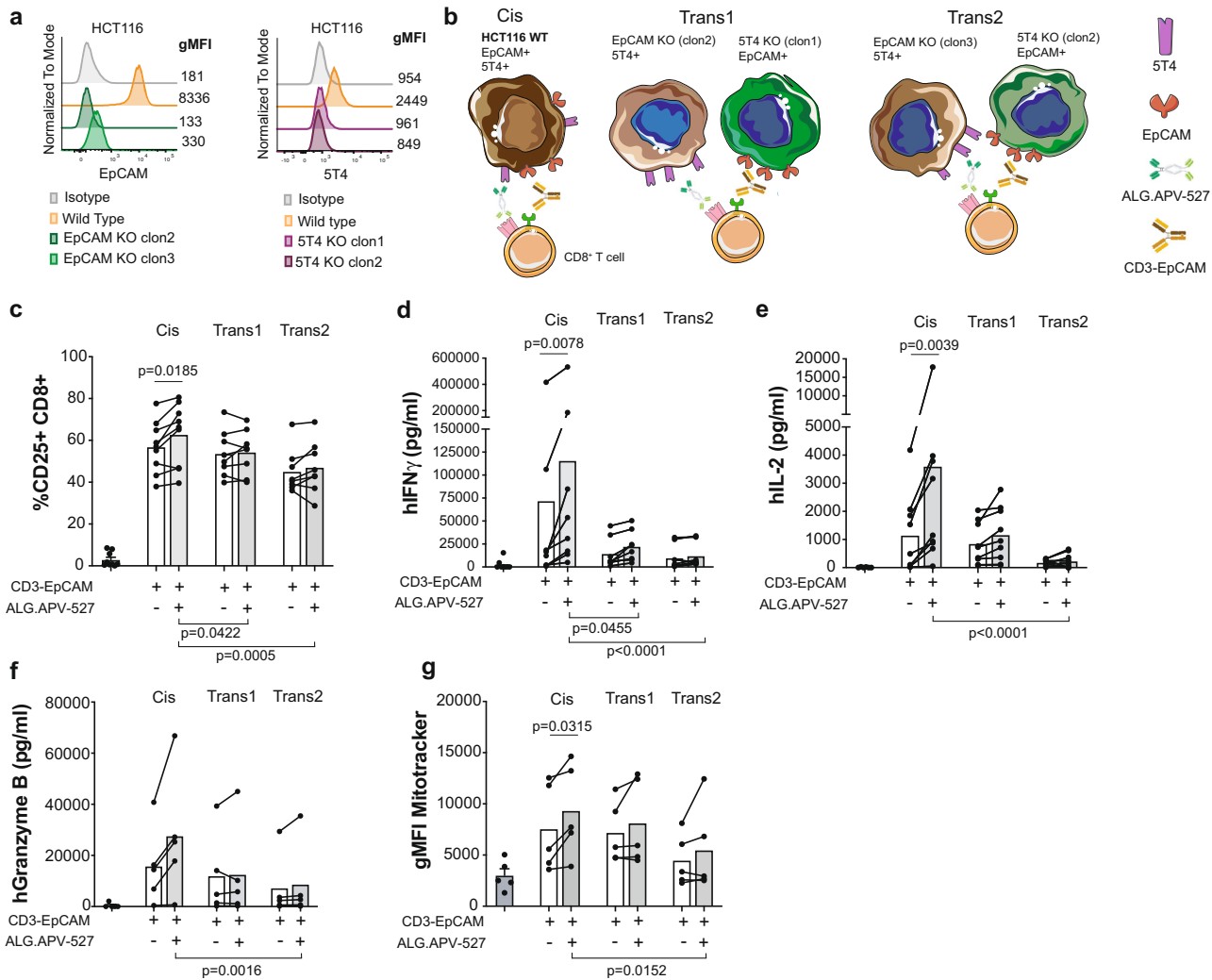

**Fig. 3 Bispecific antibodies provide cis or trans CD137-costimulation using 5T4⁺ EpCAM⁺ tumor cells or variants alternatively silenced for 5T4 or EpCAM. a**, representative histograms of EpCAM (left panel) and 5T4 (right panel) expression on wild-type HCT116 cells and on two different clones of alternatively silenced EpCAM or 5T4 HCT116 variants attained by CRISPR/Cas9 technology. Numbers indicate geometric mean fluorescence intensity, gMFI. **b**, schematic illustration of cis CD8⁺ T cell costimulation using the wild-type HCT116 cell line versus trans costimulation when plating 1:1 mixtures of the alternatively silenced variants (trans1 and trans2) in the presence of CD3-EpCAM and 5T4-CD137 (ALG.APV-527) BsAbs. **c**, cell-surface CD25 (n = 9) expression on CD8⁺ T cells analysed by flow cytometry in the indicated conditions. Concentrations of IFNγ (n = 8) (**d**), IL-2 (n = 9) (**e**) and granzyme B (n = 5) (**f**) in the co-culture supernatants from the indicated conditions. **g**, flow cytometry measurement of Mitotracker (n = 5) geometric mean fluorescence in gated CD8⁺ T cells. Statistical significance was determined with paired t tests (two-sided). Source data are provided as Source Data file.

expression of luciferase. Furthermore, experiments in co-cultures with the CD137-Jurkat reporter cell line and HCT116 cells, providing CD137-costimulation in cis versus trans upon incubation with the CD3-EpCAM and ALG.APV-527 BsAbs, also showed a more prominent NF-κB signal in the cis-costimulation conditions (Fig. 4c).

Consistent with a more pronounced NF-κB activation in cis, we observe a significant degradation of the inhibitor kappa B alpha (IκBα) in primary CD8⁺ T cells stimulated by the mAb coated beads (Fig. 4d). Moreover, p65 nuclear translocation was more prominently observed in cis versus trans by confocal microscopy (Fig. 4e) and Western blots on nuclear extracts (Fig. 4f) from primary CD8⁺ T cells. Superior activation of NF-κB upon cis-costimulation was not exclusive of the canonical NF-κB pathway, as a superior nuclear translocation of the p52 subunit of the non-canonical NF-κB pathway[31,32] was also observed in such primary human CD8⁺ T cells (Fig. 4g).

Reportedly, AKT-mTOR axis activation is also induced by CD137-costimulation[33]. In this sense, we found that

cis-costimulation provided by mAb coated beads also enhanced ribosomal S6 protein phosphorylation, an mTOR target, in primary CD8⁺ T cells in a more intense way as compared side-by-side with trans CD137-costimulation in a panel of healthy donors (Supplementary Fig. 7).

Finally, to assess if cis- versus trans costimulation could manifest in vivo in a tumor setting, HCT116-derived tumors were xenografted under the skin of immunodeficient Rag2⁻/⁻ IL2Rγc⁻/⁻ mice. To mimic cis- versus trans costimulation, either the WT or the mixtures of EpCAM and 5T4 CRISPR/Cas9-silenced variants were co-engrafted. Of note, coinjection resulted in 9-day established tumors that combined comparable percentages of each variant, thereby representing the trans costimulation conditions. CD137-Jurkat NF-κB- luciferase reporter cells were intratumorally injected together with the CD3-EpCAM and ALG.APV-527 BsAbs into such established tumors (Fig. 4h). When mice were perfused with the luciferase substrate luciferin, bioluminescence emission was examined 6 hours later, and CD137 cis-costimulation conditions rendered more photons/s/cm²/sr than in the CD137 trans

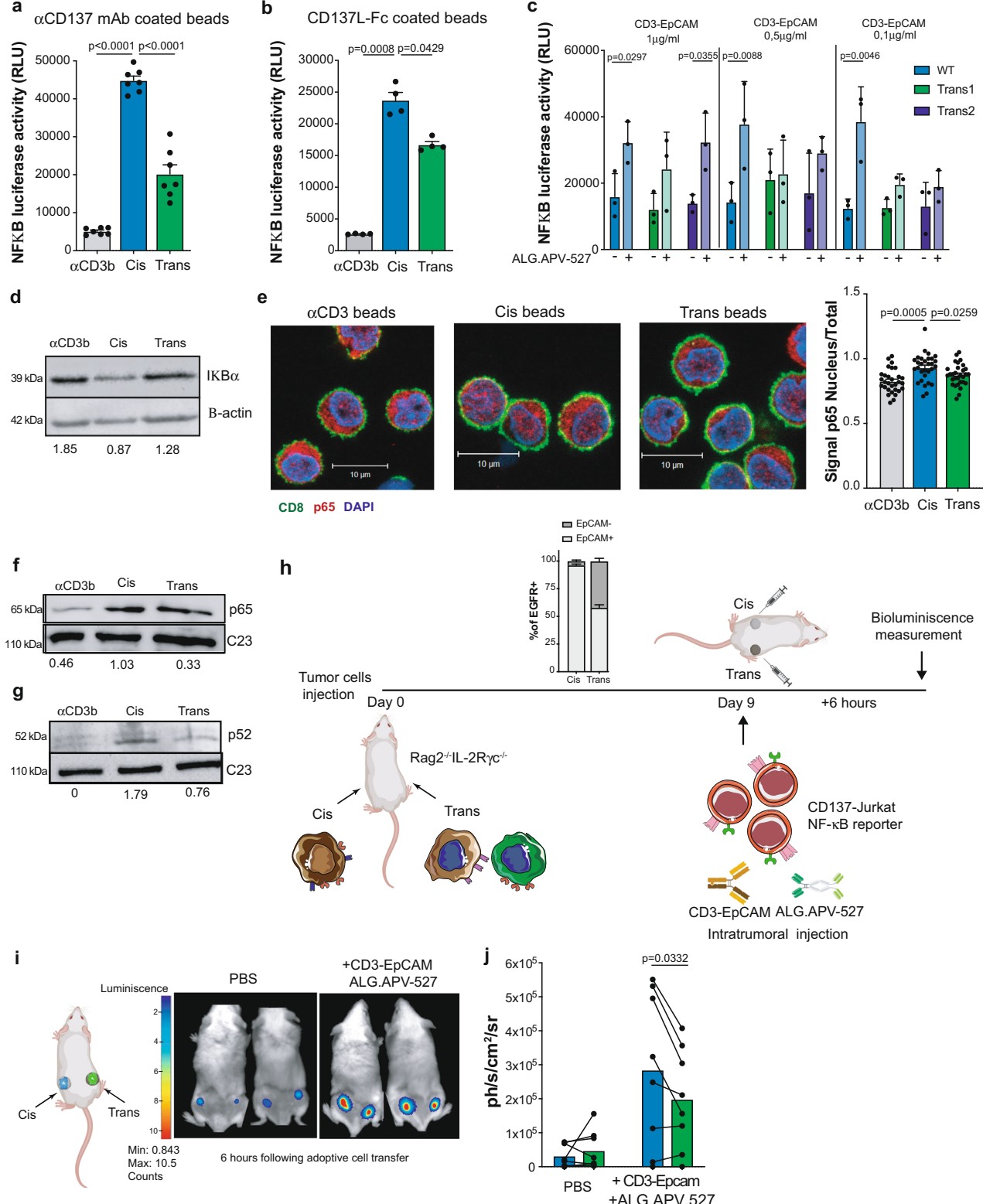

costimulation condition in vivo (Fig. 4i and j). These results indicate that the cis-costimulation induced the NF-κB pathway more intensely in the Jurkat transfected cells posing in these experiments as artificial tumor-infiltrating lymphocytes.

In summary, although we did not find qualitative differences in cis versus trans CD137-costimulation signaling, the markedly increased intensity of NF-κB activity is likely to be responsible for the observed functional differences.

## Distinct transcriptomic profiles elicited by cis CD137-costimulation.

In order to evaluate the impact of CD137 cis-versus trans costimulation at a molecular level, a genome-wide transcriptomic analyses of human CD8+ T cells from three different donors costimulated with mAb coated beads were studied by Clarion gene expression microarrays.

Interestingly, the most prominent features were the induction of genes involved in cell cycle control and replication as

**Fig. 4 Cis CD137-costimulation is superior at inducing NF-κB activation. a**, CD137-Jurkat NF-κB reporter cell line was incubated with CD137 antibody-coated microbeads ($n = 7$ technical replicates) or **b**, CD137Ligand-Fc coated microbeads ($n = 4$ technical replicates). Luciferase activity was measured after 6 h. Data are representative of at least three independent experiments. **c**, activity of CD137-Jurkat NF-κB reporter cell line was assessed by cis-costimulation with wild-type HCT116 cells or trans costimulation with combinations of the alternatively silenced variants in the presence of CD3-EpCAM and ALG.APV-527 BsAbs ($n = 3$). Data are representative of at least three independent experiments. Data are given as mean ± s.e.m. **d**, Western blot analyses of IκBα (representative of CD8+ T cells from two independent donors) on nuclear extracts from human primary human donors stimulated for 48 h with mAb coated beads. **e**, representative immunofluorescence images of p65 localization in human primary CD8+ T cells stimulated with mAb coated beads for 48 h. Cells are labeled with antibodies against p65 (red) and CD8 (green), DNA is visualized by Hoescht (scale 10 μm). Bars graph displays nuclear to total signal ratio ($n = 30$ cells each). Shown data are representative of two independent donors with similar results. Representative Western blot of p65 (**f**) and p52 (**g**) in nuclear extracts of human primary human donors stimulated for 48 h with mAb coated beads. Numbers below Western blot indicate relative expression normalized to the total expression of β-actin for **d** and to C23 for **f** and **g** as analysed by densitometry. Data are representative of at least two independent donors. **h**, schematic layout of the experiment in **i** and **j**. Wild-type HCT116 cells and a mixture of 1:1 ratio of HCT116 KO EpCAM (clon3) together with HCT116 KO 5T4 (clon1) were subcutaneously engrafted in the left flank or in the right flank of Rag2−/− IL-2Rγc−/−, respectively. Tumors were established for 9 days before intratumoral injection of CD137-Jurkat-4 NF-κB reporter cell line co-injected with CD3-EpCAM and ALG.APV-527 BsAbs. The relative presence of the tumor cell clones in the xenografted tumors are shown in an inset in the experimental layout. Photon emission was measured with a PhotonIMAGER from the anesthetized living animals. In vivo imaging (**i**) and quantification (**j**) ($n = 8$) of luciferase activity from CD137-Jurkat NF-κB reporter cell line intratumorally transferred into cis or trans HCT116 paired tumors of each mouse in the presence of CD3-EpCAM and ALG.APV-527 BsAbs 6 h postreporter cell injection. Summary data are given as the bioluminescence photon flux subtracted from the tumors prior to the experiment (background). As negative controls mice whose tumors were injected with the reporter CD137-Jurkat transfectants and PBS are shown. Each pair of dots represents a single mouse bearing a cis tumor and a trans tumor in the left and right flank, respectively. RLU relative luciferase units. Scale bars: 10 μM. In **d** and **f-g** samples derive from the same experiment and loading control gels were processed in parallel. Statistical significance was determined with one-way Anova with Tukey's multiple comparison test for **a**, **b** and **e** (one-sided), the Friedman test with Dunn's correction for **c** and paired t test for **j** (two-sided). Source data are provided as Source Data file.

well as in DNA damage repair (Fig. 5a–c). To validate these observations, we studied if human CD8+ T cells costimulated in cis by mAb coated beads proliferated more avidly through the analysis of Ki67 expression (Fig. 5d) and by a violet dye-based proliferation assay (Fig. 5e). In accordance to the transcriptomic analysis, a more intense proliferation was observed as a result of cis-costimulation compared to trans costimulation conditions. Indeed, more intense proliferation was observed comparing cis versus trans conditions and more DNA synthesis and replication was found assessing EdU incorporation into genomic DNA after cell co-cultures of CD8+ T cells with the WT HCT116 cell line and silenced variants combinations in the presence of the CD3-EpCAM and ALG.APV-527 BsAbs (Fig. 5f).

Importantly, less signs of DNA damage were observed in proliferating CD8+ T cells when costimulated in cis as detected by levels of phosphorylated gH2AX histone (Fig. 5g). Moreover, each sequential generation of dividing cells showed increasing levels of phosphorylated gH2AX staining after CD137-costimulation in trans, whereas DNA damage accumulation was less intense in CD8+ T cells that had received CD137-costimulation in cis (Fig. 5h).

These results indicate that cis versus trans CD137-costimulation provided replicative advantages, while preserving the integrity of genomic DNA undergoing replicative stress during clonal expansion.

**Mouse CD8+ T cells show superiority of cis CD137-costimulation.** To examine if cis- was superior to trans costimulation in mouse CD8+ splenocytes, CD8+ T cells were exposed to microbeads coated jointly or separately with anti-mouse CD3 and/or antimouse CD137 mAbs (Supplementary Fig. 8a). We found an advantage of cis over trans CD137-costimulation in terms of CD25, PD-1 and Ki67 expression (Supplementary Fig. 8b–e). These experiments conclude that cis-costimulation superiority is conserved between human and mouse. Furthermore, we compared the cytotoxicity of OT-1 cells in culture against B16.OVA cells. Supplementary Fig. 8f shows that cis preactivated OT-1 effectors performed more intense

cytotoxicity as compared to those OT-1 lymphocytes that had been costimulated in trans.

Making use of the differential levels of cis- versus trans costimulation in mouse CD8+ T cells, mixtures of 1:1 cis-costimulated CD8+ T cells (CD45.1) and trans-costimulated CD8+ T cells (CD45.2) were adoptively transferred into Rag1−/− recipient mice that lack endogenous T cells (Fig. 6a). Follow-up of adoptively transferred T cells over time indicated that cis-costimulated lymphocytes dominated over trans-costimulated lymphocytes in terms of numbers as late as 12 and 20 days following adoptive transfer (Fig. 6b). These findings were recapitulated in T cells present in liver, spleen, lymph nodes and bone marrow when mice were sacrificed twenty days after competitive adoptive transfer (Fig. 6c).

To address potential mechanisms, we differentially studied CD45.1 and CD45.2 gated T lymphocytes by FACS to ascertain the accumulated levels of DNA damage. Clear differences suggesting less accumulation of DNA damage in cis CD137-costimulated cells was found in lymphocyte suspensions from the different organs (Fig. 6d). Furthermore, analysis of expression of T-bet and Eomes transcription factors in the bone marrow suspensions clearly showed enhanced expression of T-bet in the CD8+ cells that had received cis CD137-costimulation (Fig. 6e±g). These intriguing results suggest that cis-costimulation during priming functionally reprograms T cells and protect them from replication stress.

Next, we sought to perform experiments of competitive infiltration of in vivo engrafted B16.OVA melanomas following adoptive transfer of OT-1 cells prestimulated with beads providing cis- (CD45.1+) or trans costimulation (CD45.2+). Experiments as those graphically described in Fig. 6h showed that resulting tumors and tumor-draining lymph nodes (dTLN) were more abundantly infiltrated or populated by OT-1 cells that had been CD137-costimulated in cis (Fig. 6i).

**Vaccination with CD137L-OVA-expressing vectors in cis are superior.** To study the in vivo relevance of the observed superiority of cis over trans CD137-costimulation, we sought to immunize mice with previously published OVA encoding

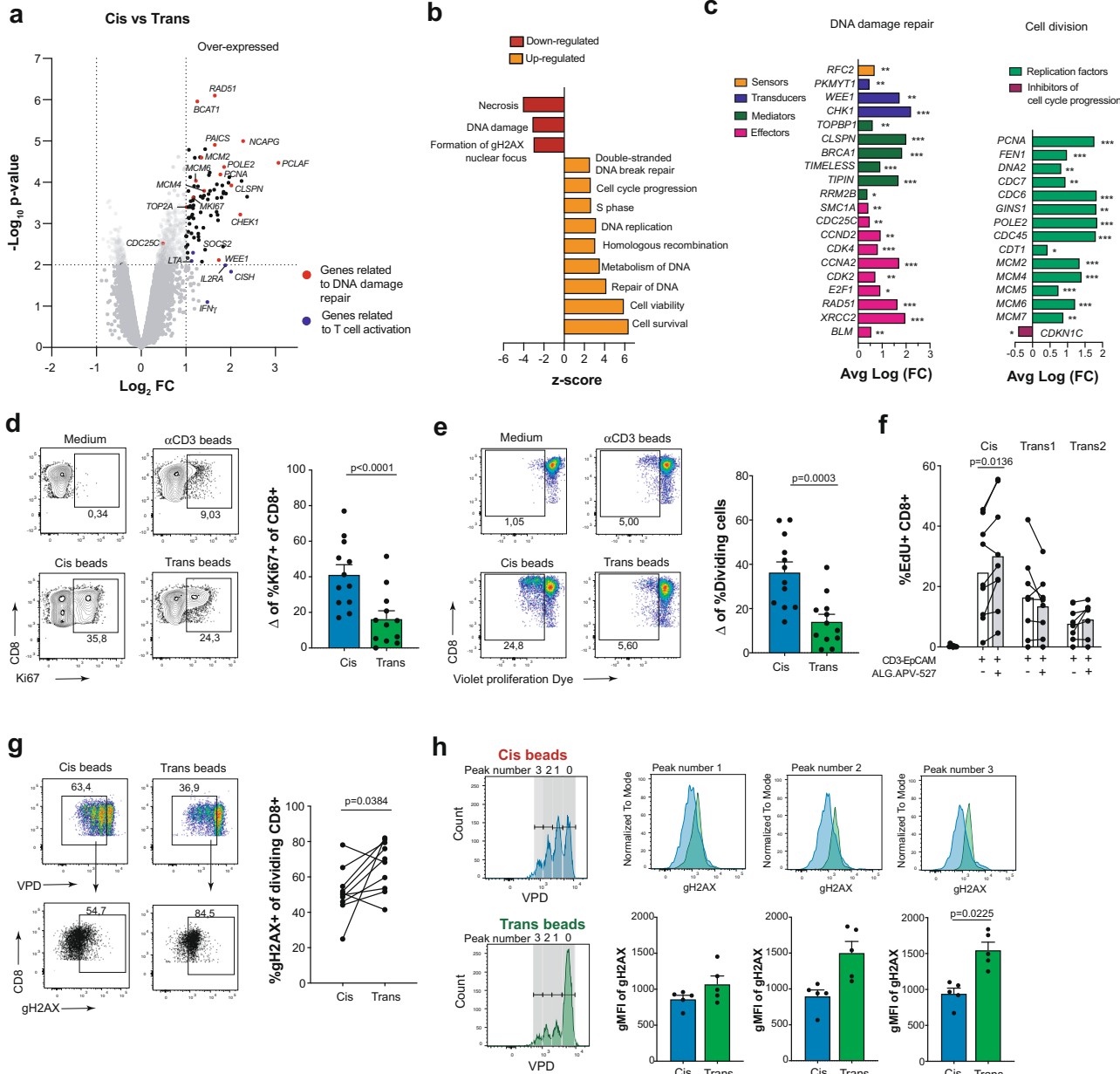

**Fig. 5 Transcriptomic differences of cis versus trans CD137 costimulation have implications in T cell proliferation and DNA repair. a**, volcano plot of differentially expressed genes of cis-costimulated and trans-costimulated CD8+ T cells with mAb coated microbeads for 32 hours. Red dots indicate genes related to DNA damage/repair and blue dots indicate genes related to T cell activation. Dotted lines mark the *t* test *p* value cut-off of p < 0.01 and log₂FC > 2. **b**, Top significantly affected pathways based on Ingenuity pathway analysis (IPA). The horizontal bars denote the different pathways based on the z-score. Orange color indicates upregulation, while red color indicates downregulation. **c**, top genes displaying altered expression in the DNA damage/repair pathway (left panel) and cell division (right panel) in cis- versus trans costimulation. **d**, representative dot plots of nuclear Ki67 expression in CD8+ T cells after costimulation with mAb coated beads. Cumulative data (n = 12) are shown as the difference between the value of cis and trans conditions to which the anti-CD3 background was subtracted in each case. **e**, CD8+ T cells were labeled with a violet proliferation dye (VPD) and stimulated with mAb coated beads for 96 h. Representative dot plot of proliferating CD8+ T cells and cumulative data (n = 12). Cumulative data are shown as the difference between the value of cis and trans conditions to which the anti-CD3 background was subtracted in each case. **f**, DNA synthesis was assessed in CD8+ T cells by cis-costimulation with wild-type HCT116 cells or trans costimulation with the alternatively EpCAM and 5T4 silenced variant combinations in the presence of CD3-EpCAM and ALG.APV-527 BsAbs (n = 9 for cis, n = 8 for trans1 and trans2). Co-cultures were treated with 10 μM EdU for 2 h and its incorporation to genomic DNA was detected according to the recommended staining protocol. VPD labeled CD8+ T cells were analyzed for DNA damage after stimulation with mAb coated beads. Intranuclear staining of phosphorylated gH2AX on total proliferating CD8+ T cells (**g**) (n = 10) and cells analyzed as per division cycle upon gating (**h**) (n = 5) was measured by flow cytometry. The division peaks are numbered from 1 to 3. Data are given as mean ± s.e.m. Statistical significance was determined with paired t test (two-sided) for **d**, **e**, **g** and **h** and one-way Anova (one-sided) with Holm-Sidak multiple comparison test for **f**. (*p < 0.05, **p < 0.01, ***p < 0.001). Source data can be retrieved under accession number GSE158041 and as Source Data file.

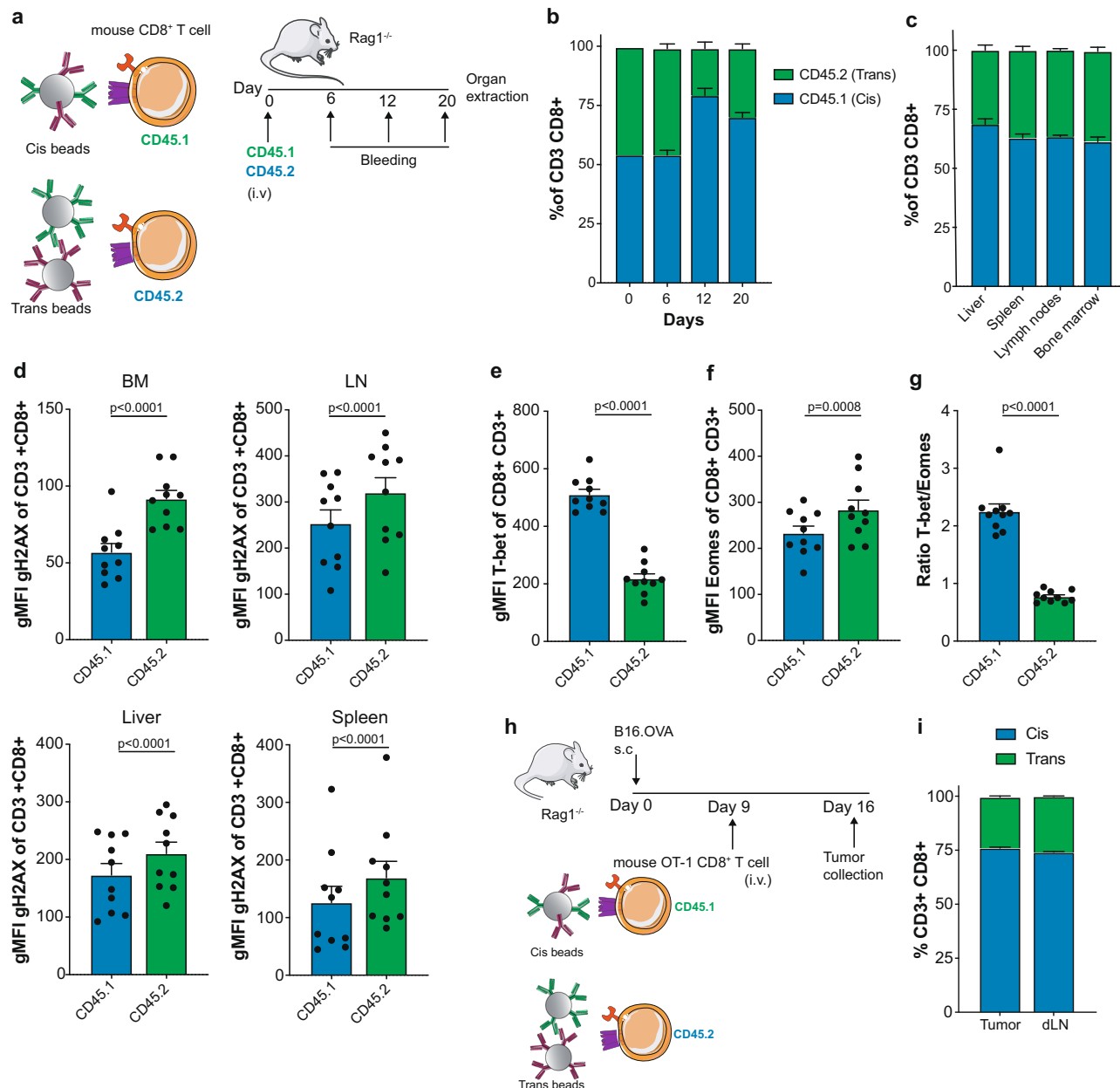

**Fig. 6 In vivo consequences of cis versus trans CD137 costimulation. a**, schematic layout of the experiments in **b**–**g**. A mixture of 1:1 ratio of in vitro cis-costimulated (CD45.1) and trans-co-stimulated (CD45.2) CD8$^+$ T cells from C57 mice were adoptively transferred upon intravenous injection (i.v) into Rag1$^{-/-}$ congenic recipients. **b**, frequency of CD45.1 and CD45.2 of CD3$^+$CD8$^+$ T cells in the blood on days 6, 12 and 20 post-adoptive transfer ($n = 10$). **c**, frequency of CD45.1 and CD45.2 of CD3$^+$CD8$^+$ T cells in the liver, spleen, lymph nodes and bone marrow on day 20 post-adoptive transfer ($n = 10$). **d**, intranuclear staining of phosphorylated gH2AX on CD45.1 and CD45.2 FACS-gated CD3$^+$CD8$^+$ in the liver, spleen, lymph nodes (LN) and bone marrow (BM) on day 20 post-adoptive cell transfer ($n = 10$). Data show gMFI. Each dot represents a single mouse. Expression of T-bet (**e**) and Eomes (**f**) and T-bet:Eomes ratio (**g**) in CD45.1 and CD45.2 FACS-gated CD3$^+$CD8$^+$ in the bone marrow on day 20 post-adoptive cell transfer ($n = 10$). Data show geometric mean fluorescence intensity (gMFI). Each dot represents a single mouse. **h**, experimental layout of the experiment in i. A mixture of 1:1 ratio of cis-costimulated (OT-1 CD45.1$^+$) and trans-co-stimulated (OT-1 CD45.2$^+$) CD8$^+$ T cells were adoptively transferred into subcutaneous (s.c) implanted B16.OVA tumor-bearing Rag1$^{-/-}$ recipient mice. **i**, relative frequency of OT-1 CD45.1$^+$ and OT-1 CD45.2$^+$ CD8$^+$ T cells in the tumor and draining lymph nodes (dLN) on day 16 post-adoptive transfer ($n = 10$). Data are given as mean ± s.e.m. Statistical significance was determined with paired t test (two-sided). Source data are provided as Source Data file.

Modified Vaccinia Ankara (MVA) vectors that encode CD137L (MVA-OVA-4-1BBL) in the same vector or when it is provided by separate vectors (MVA-OVA + MVA-4-1BBL)[34]. With these agents, we intravenously immunized mice in a prime/boost scheme shown in Fig. 7a. OVA-specific CD8$^+$ T cells were traced in blood over time, while mice received prime and boost regimens with the MVA vectors. Fig. 7b shows that OVA-specific T cells

identified with proper MHC-I multimers were expanded more avidly by the virus providing CD137L costimulation in cis than the combination of viruses providing CD137L costimulation in trans. These differences were more striking following boost intravenous inoculations. Such results were also found in CD8$^+$ splenocytes at the end of the experiments (Fig. 7c) and such splenocytes accumulated less DNA damage as shown by weaker

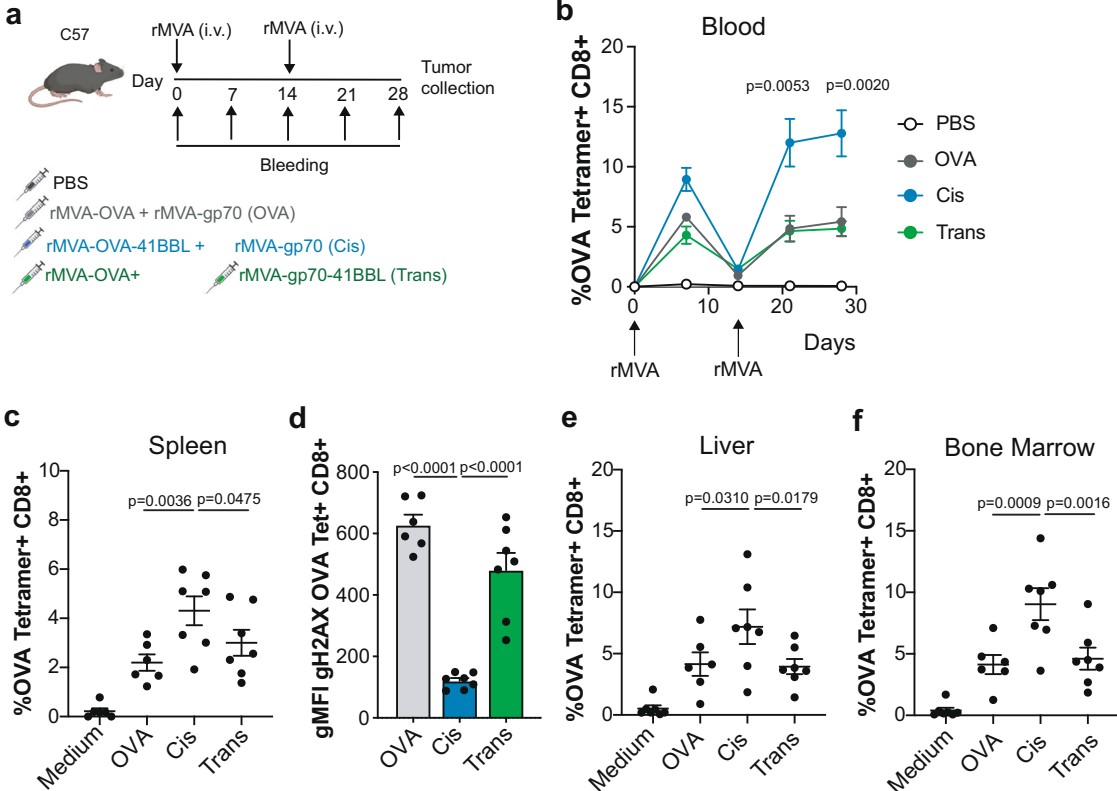

**Fig. 7 MVA vector immunizations providing CD137L costimulation in cis are superior at priming and boosting anti-OVA CTLs. a** schematic layout using MVA vectors encoding both OVA and CD137L or each transgene in a separate fashion. C57/Bl6 mice were immunized with $5 \times 10^7$ TCID$_{50}$ of the different rMVAs at day 0 and day 14 ($n = 7$ for PBS, $n = 6$ for rMVA-OVA, $n = 7$ for cis-costimulation and $n = 7$ for trans costimulation). **b**, OVA-specific CD8$^+$ T cell response was quantified by tetramer staining of peripheral blood CD8$^+$ T cells at the indicated timepoints. **c**, OVA-specific CD8$^+$ T cells in the spleen were determined by H-2K$^b$ SIINFEKL tetramer following priming and boost intravenous inoculations. **d**, intranuclear staining of phosphorylated gH2AX on OVA-specific CD8$^+$ T cells from spleens that was measured by flow cytometry. Frequency of OVA-specific CD8$^+$ T cells measured on the cell suspensions from liver (**e**) and bone marrow (**f**) on day 28 after prime/boost with the different rMVAs. OVA, ovoalbumin. i.v., intravenous. Data are given as mean ± s.e.m. Statistical significance was determined with one-way Anova (one-sided) with Holm-Sidak multiple comparison test. Source data are provided as Source Data file.

gH2AX immunostaining in the cis condition (Fig. 7d). Such observations of more efficacious anti-OVA immunization by MVA-OVA-4-1BBL versus MVA-OVA plus MVA-4-1BBL were also made in cell suspensions derived from the liver and bone marrow (Fig. 7e, f). These experiments conclude that immunization for CTL generation is more efficient if CD137L is provided in cis with regard to the antigen-expressing infected cell.

## Discussion

Our work shows that CD8$^+$ costimulation with CD137 agonists in cis versus trans with respect to the antigen source is functionally superior both in human and mouse experimental systems. These findings are highly relevant for the development of immunotherapeutic agents targeting CD137[5].

The intrinsic mechanisms of signaling indicate a potential crosstalk of CD3-TCR signaling and CD137 signaling when co-ligated at the same immune synapse[35]. Previous confocal microscopy observations have suggested a function for mouse CD137 costimulation in organizing CD3 signaling at the immune synapse[36]. We observed in this study a marked enhancement of NF-κB signals in the cis-costimulation conditions. NF-κB is a signaling route that is reached both from CD3-elicited MALT1/CARMA/BCL10 complexes[37] and CD137 acting via TRAF2[38], both pathways converge on the IKK complex, which can be the hub in the signaling crosstalk between TCR-CD3 and CD137. However, the molecular mechanisms behind the crosstalk of these

receptors, which result in such synergistic signaling outcomes in a cell polarized manner, remain to be seen. Our transcriptomic observations show that genomic DNA preservation and resistance to replicative stress could at least in part account for cis-costimulation superiority, which in our hands leads to a competitive advantage in vivo following adoptive transfer. T-bet/Eomes transcriptional regulation could also have a function. Examples indicating superiory of costimulation in-cis are offered by the CD28-B7s pathway[39] and similar behaviours might be postulated to occur in the case of other costimulatory members of the TNFR family, such as for instance CD27, OX40, CD30 and GITR[40].

The most striking finding is that provision of CD137-costimulation in cis gives rise not only to more avid proliferation, but importantly to protection against DNA damage[41], which is prominent in T cells undergoing activation and clonal expansion[42]. Such mechanisms, in conjunction with Bcl-xL expression, more fit mitochondria, and IL-2 autocrine production, provide cis CD137-costimulated T cells with clear competitive survival and functional advantages. This advantage was prominently observed upon competition to infiltrate transplanted tumors and in response to vaccination with MVA vectors[34,43]. The intriguing effects of Eomes and T-bet expression are probably behind these observations, but such transcriptional biology is very complex and gives rise to different outcomes depending on balanced coexpression of these two transcription factors[44].

When considering natural antigen presentation by mature dendritic cells, it can be envisioned that those presenting cognate antigen would provide costimulation in cis, whereas those presenting irrelevant antigens would provide costimulation in a trans-like fashion. Physiologically, this would favor T cells receiving cis-costimulation as a safety and antigen-focussing feature built into the system that controls potentially dangerous specific T cell cytotoxicity. The intimate difference between cis and trans CD137-costimulation might involve differential regulation of several elements in immune synapses including the cytoskeleton and crosstalks with other receptors that remain to be studied. Mitochondrial redistribution and 4-1BB relocalization to immune synapses have been previously reported[36,45], but in our hands is not different in the cis versus trans conditions. Of note, the superiority of cis 4-1BB costimulation was also substantiated in human CD8+ T cells previously primed under the influence of CD28 to resemble more physiological conditions of T cell activation.

These results altogether are of importance in cancer immunotherapy, as agents are being developed to provide CD137-costimulation in cis or in trans with respect to the cells presenting tumor antigens[5,19,21]. Although our results would favor cis versus trans, it remains to be seen if the intensity of CD137 ligation could compensate for its provision in trans[46,47]. The most extreme example of successful CD137 in-cis ligation is provided by CAR T cells that have revolutionized the treatment of B-cell malignancies[24,25]. These constructs incorporate the cytoplasmic tail of CD137 in their CARs, thereby delivering artificial in-cis costimulation. The intricacies of cis- versus trans costimulation offer translational clues to clinically exploit CD137-targeting agents for immunotherapy of cancer.

## Methods

**Human subjects and sample collection**. This study was approved by the Clinica Universidad de Navarra Ethics Committee (2019-039) and Navarra Blood and Tissue Bank Navarrabiomed Biobank. Written informed consent was obtained from all healthy and unrelated participants. We have complied with all relevant ethical regulations for work with human participants, in accordance with the Declaration of Helsinki. A total of 50 caucasian volunteers participated in the study.

**Mice**. Experiments involving mice were approved by the Ethics Committee of the University of Navarra (R-030-19GN). We have complied with all relevant ethical regulations for animal testing and research. C57BL/6 mice (5-6 weeks old, male) were obtained from Envigo (Huntingdon, Cambridgeshire, UK) and maintained in the animal facility of Cima-Universidad de Navarra. Wild-type C57 BL/6, CD45.1+ (C57 BL/6), OT-1 (C57 BL/6), OT-1 CD45.1+ (C57 BL/6), B6.129S7-$Rag1^{tm1Mom}$/J (C57 BL/6 Rag1) and C;129S4-$Rag2^{tm1.1Flv}$ $Il2ry^{tm1.1Flv}$/J (C57 BL/6 Rag2$^{-/-}$ IL2Rγc$^{-/-}$) mice were bred at CIMA Universidad de Navarra in specific pathogen-free conditions.

**Cells line and reagents**. HCT116 cell line was obtained from ATCC and it was maintained at 37°C in 5% CO$_2$ and grown in complete RMPI (cRPMI); RPMI Medium 1640+Glutamax (Gibco, Invitrogen, Carlsbad, CA) containing 10% heat-inactivated FBS (SIGMA,), 50 IU/mL penicillin and 50 µg/mL streptomycin (Gibco). The B16-OVA cells were provided by Dr. Lieping Chen (Yale University, New Haven, CT) and were grown in cRPMI supplemented with 1% of B-mercaptoethanol 55 mM (Gibco).

The BsAb targeting CD3 and EpCAM was generated using controlled Fab-arm exchange[48] using the variable sequences of OKT3 (Orthoclone) to target CD3 and of solitumab (MT110) to target EpCAM. The ALG.APV-527 molecule is a human bispecific antibody, that comprises a hinge-Fc from human IgG1, and human scFvs targeting both 5T4 or CD137. GlySer linkers were used in the scFv and to attach the scFv targeting CD137 to the COOH-terminal Fc portion of the IgG1 molecule (in preparation). The Fc portion was mutated to remove binding to Fc-gamma receptors and remove complement fixation. These mutations are required to ensure target-dependent function of ALG.APV-527 and prevent Fc-mediated clustering of CD137.

**Cell isolation**. PBMCs were isolated by density gradient (Ficoll–Hypaque, Amersham Biosciences) from heparinized blood of healthy donors (n = 50). CD8+ T cells were freshly isolated by a negative magnetic selection kit (CD8+ T cell isolation kit, 130-094-156, Miltenyi Biotec). For microarray analysis, high-purity (>95%) CD8+ T was required.

Spleens from naïve C57/B6, OT-1 and OT-1 CD45.1 mice were excised and splenocytes suspensions were mechanically prepared. CD8+ T cells were isolated by a negative magnetic selection with the CD8+ T cell isolation kit (130-094-156, Miltenyi Biotec).

**Microbead coating and preparation**. Dynabeads M-450 Tosylactivated (Invitrogen) were used to covalently couple monoclonal antibodies. Briefly, beads were incubated in 1 ml of phosphate buffer 0.1 M pH 7.6 with each mAb at 37°C overnight in rotation. Unreacted tosyl groups were blocked by incubation with buffer tris 0.2 M 0.1%BSA pH 8.5 for 30 min at room temperature. Beads were then washed twice with 1% bovine serum albumin (BSA) in phosphate-buffered saline (PBS) 2 mM EDTA pH 7.4 and stored at 4°C. The following monoclonal antibodies were used to couple microbeads to simulate human T cells: anti-human CD3 (BE0001-2, Bio X cell) and anti-human CD3 (produced in house), anti-human CD137 (6B4, mouse IgG1, produced in house), mouse IgG2a (40202, Biolegend), mouse IgG1 (400102, Biolegend) (Supplementary Table 1). In the case of microbeads coupled to CD137L, the CD137L-Fc (Sino Biological) was used (Supplementary Table 2). In the case of microbeads coupled to anti-CD7, the anti-CD7 mAb (395602, Biolegend) was used. The following monoclonal antibodies were used for mAb coated beads to stimulate mouse T cells (Supplementary Table 3): antimouse CD3 (100238, Biolegend), antimouse CD137 (BE0239, Bio X cell), antimouse CD137 (3H3, produced in house), rat IgG2a (400533, Biolegend), rat IgG2b (400637, Biolegend). Details of concentrations for antibody coupling are provided in the tables.

**CD8+ T cell activation**. CD8+ T cells were magnetically sorted by negative selection either from human PBMCs or from the spleens of naïve mice. CD8+ T cells were stimulated in 200 µl of cRPMI with 3:1 beads:T cells ratio for 96 hours. Different bead:T cell ratios were used when indicated. Trans beads were a mixture of 1:1 ratio of anti-CD3 coated beads and anti-CD137 coated microbeads. 1 µg/ml of plate-bound anti-CD3 mAb (OKT3, produced in house) and 2 µg/ml of soluble anti-CD28 mAb (102115, Biolegend) were used to pre-activate CD8+ T cells for 16 h.

**Human T cell proliferation assays**. To determine CD8+ T cells proliferation upon co-culture with mAb coated microbeads, CD8+ T cells were magnetically sorted out from human PBMCs and activated for 96 hours. Prior to plating the co-culture experiment, CD8+ T cells were stained with CellTrace Violet Cell Proliferation Kit (C34571, Thermo Fisher) following the manufacturer's instructions. Cell proliferation was monitored by Violet dye staining dilution in CD8+ T cell population by flow cytometry.

For the assessment of replicating DNA of CD8+ T cells, $12.5 \times 10^4$ wild-type HCT116 cells and a mixture of CRISPR/Cas9-silenced variants were seeded in a 96-well plate. After overnight culture, primary CD8+ T cells were added in a 5:1 effector:target ratio in the presence of 0.1 µg/ml CD3-EpCAM and 1.5 µg/ml of ALG.APV-527 BsAbs. Following 48 hours of co-culture, CD8+ T cells were stained for flow cytometry analysis. Following 46 hours of co-culture, 10 mM of 5-ethynyl-2'-deoxyuridine (EdU, Invitrogen) was added for 2 hours. CD8+ T cells were then surface stained and EdU incorporation was assessed with the Click-iT Plus EdU Alexa Fluor 488 Flow Cytometry Assay Kit (C10366, Invitrogen). All samples were acquired on a BD Canto II. All analyses were performed using Flowjo software (Tree Star).

**Flow cytometry**. For flow cytometry studies, dead cells were excluded using a Zombie NIR Fixable Viability kit (1:1000, 423106, BioLegend) or PKPF (1:1000 840301, PromoCell). Human cells were surface stained with the following fluorochrome-labeled antibodies purchased from BioLegend: anti-CD8-BV510 (1:200, 344732), anti-CD25-APC (1:200, 302610), anti-CD45-PE-Cy7 (1:200, 304016), anti-PD-1-PerCPCy5.5 (1:200, 329914). For intracellular staining cells were permeabilized after surface staining with Cytofix/Cytoperm (BD) for 10 min following the manufacturer's instructions and intracellularly stained with anti-Bcl-xL-PE (1:200, 10030-09, Southern Biotech). For intranuclear staining, cells were fixed and permeabilized after surface staining with True nuclear Transcription Buffer Set (424401, Biolegend) for 30 min following the manufacturer's instructions and intracellularly stained with Ki67-AF488 (1:200, 350508, Biolegend), T-bet-BV421 (1:200, 644816, Biolegend) and Eomes-PE (1:200, 12-4877-42, eBioscience).

For intranuclear staining of phospho-S6, dead cells were excluded using a Zombie NIR Fixable Viability kit (1:1000, 423106, BioLegend). Cells were stained for surface markers and then fixed with 4% of paraformaldehyde. Cells were then permeabilized with cold Perm III Buffer (558050, BD). Then, anti-phospho-S6-AF647 (4851, Cell Signaling) was detected by intranuclear staining.

Mouse T cells were surface stained with the following fluorochrome-labeled antibodies purchased from Biolegend: anti-CD3-AF647 (1:200, 100209), anti-CD8-BV510 (1:200, 100752), anti-CD19-BV650 (1:200, 115541), anti-CD25-FITC (1:200, 102006), anti-CD25-APC (1:200, 102012), anti-CD45.2-FITC (1:300, 109806), anti-CD45.1-PE (1:300, 110708), anti-CD45.2-Pacific Blue (1:300, 109820), anti-PD-1-PerCPCy5.5 (1:200, 135208) and anti-CD3-BUV496 (1:100,

612955, BDBioscience). For intranuclear staining, cells were fixed and permeabilized after surface staining with True nuclear Transcription Buffer Set (424401, Biolegend) for 30 min following the manufacturer's instructions and intracellularly stained with Ki67-AF700 (1:200, 652420, Biolegend), T-bet-PE-Cy7 (1:200, 644816, Biolegend) and Eomes-eFluor450 (1:200, 48-4875-82, eBioscience). Antigen-specific T cells were detected from cell suspensions after staining with H-2K$^b$ OVA PE (1:200, TB-5001-1, iTag MHC Tetramer, MBL International, Woburn, MA, USA) and True stain FcX antimouse CD16/32 (1:100, 101320, Biolegend) for 15 minutes at 4 °C.

HCT116 cell line was surface stained with the following antibodies: anti-EpCAM-PerCPCy5.5 (324214, Biolegend) and anti-5T4 (Alligator Bioscience).

For intracellular staining of gH2AX, dead cells were excluded using a Zombie NIR Fixable Viability kit (1:1000, 423106, BioLegend) or PKPF (1:10000 840301, PromoCell). Cells were stained for surface markers and then fixed with 4% of paraformaldehyde. Cells were then permeabilized with cold Perm III Buffer (558050, BD). Then, gH2AX-PE (1:200, 613412, Biolegend) or gH2AX-AF647 (1:200, 560447, BD Bioscience) was detected by intranuclear staining. For gH2AX assessment on proliferating human CD8$^+$ T cells, prior to establishing the co-culture, CD8$^+$ T cells were stained with CellTrace Violet Cell Proliferation Kit (1 μM, C34571, Thermo Fisher) following the manufacturer's instructions. All samples were acquired on a BD Canto II, CytoflexS or CytoflexLX. All analyses were performed using Flowjo software v.10.6.2 (Tree Star) or Cytexpert.

Antibody-coated microbeads were stained with antimouse IgG1-PE (1:200, 406608, Biolegend), antimouse IgG2a-AF647 (1:200, A21241, Invitrogen), anti-rat IgG2a-AF647(1:200, ab172333, abcam), anti-rat IgG2b AF488 (1:200, ab172334, abcam).

Gating strategies are shown in Supplementary Fig. 9.

**Mitotracker and TMRM measurement.** Mitochondrial mass and membrane potential were assessed by flow cytometry using the MitoTracker deep red (10 nmol/L, M22426, Invitrogen) and the potentiometric dye tetra-methylrhodamine methyl ester (TMRM) (125 ng/ml, T668, Invitrogen) in complete culture media for 20 minutes at 37 °C. Cells were then stained for surface markers. All samples were acquired on a BD Canto II. All analyses were performed using Flowjo software (Tree Star).

**CD137 costimulation with ALG.APV-527 bispecific antibody.** Wild-type HCT116 cells were seeded at $5 \times 10^4$ into a 96-well plate. After overnight culture, $2.5 \times 10^5$ human primary CD8$^+$ T cells were added in a 5:1 effector:target ratio in the presence of 1, 0.5 and 0.1 μg/ml CD3-EpCAM and 1.5 μg/ml of ALG.APV-527 BsAbs. Following 48 hours of co-culture, CD8$^+$ T cells were stained for flow cytometry analysis.

For the assessment of CD137 cis- versus trans costimulation, $12.5 \times 10^4$ wild-type HCT116 cells or a mixture of CRISPR/Cas9-silenced variants were seeded in 96-well plates. Trans1 is referred to the 1:1 ratio combination of HCT116 5T4$^+$ EpCAM$^-$ clon2 with HCT116 5T4$^-$ EpCAM$^+$ clon1 cells, while trans2 is referred to the 1:1 ratio combination of HCT116 5T4$^+$ EpCAM$^-$ clon3 with HCT116 5T4$^-$ EpCAM$^+$ clon2 cells. After overnight culture, $1.25 \times 10^5$ human primary CD8$^+$ T cells were added in a 5:1 effector:target ratio in the presence of previously optimized 0.1 μg/ml CD3-EpCAM and 1.5 μg/ml of CD137-5T4 BsAb. Following 48 hours of co-culture, CD8$^+$ T cells were stained for flow cytometry analysis.

**Cytotoxicity assay.** Cytotoxic killing of target cells was assessed using the xCELLigence Real-Time Cell Analyzer System (ACEA Biosciences). B16.OVA target tumor cells were plated (1,5 × 10$^4$ cells/well). After 4 hours cell adherence, 8 hours cis- and trans-stimulated OT-1 CD8$^+$ T cells were added at a 5:1 ratio (Effector:Target). Cell index was monitored every 5 minutes for 40 hours and normalized to the maximum cell index value immediately prior to effector-cell plating.

**In vivo CD137-costimulated CD8$^+$ T cells.** Isolated mouse CD8$^+$ T cells from the spleen of CD45.1$^+$ and CD45.2$^+$ C57BL/6 mice were stimulated with mAb coated beads for 48 hours. CD8$^+$CD45.1$^+$ T cells were activated with cis mAb coated microbeads, whereas CD8$^+$CD45.2$^+$ T cells were activated with trans mAb coated microbeads (microbeads coated with anti-CD3 plus microbeads coated with anti-CD137). A mixture of $3 \times 10^6$ of CD45.1$^+$ and $3 \times 10^6$ CD45.2$^+$ CD8$^+$ T cells were intravenously injected in 50 μL of saline in Rag1$^{-/-}$ mice.

**Immunization of mice.** Modified virus Ankara constructs (MVA-BN) were developed by Bavarian Nordic[34]. Bavarian Nordic kindly provided the following recombinant MVAs encoding for ovoalbumin (MVA-OVA), gp70 (MVA-gp70), ovoalbumin and 4-1BBL (MVA-OVA-4-1BBL) and gp70 with 4-1BBL (MVA-gp70-4-1BBL). A group of mice received rMVA-OVA in combination with rMVA-gp70-4-1BBL (Trans). Another group of mice received rMVA-gp70 in combination with rMVA-OVA-4-1BBL (Cis). MVA constructs were administered a total volume of 200 μl containing $5 \times 10^7$ TCID$_{50}$ of the respective MVA recombinants. The left lateral tail vein was used for intravenous injection of rMVAs encoding for tumor-associated antigens (TAA), whereas the right lateral vein was used for intravenous injection of rMVAs encoding for TAA-4-1BBL.

**Tumor cell injection.** Rag1$^{-/-}$ mice were injected subcutaneously in the flank with $5 \times 10^5$ B16.OVA tumor cells. On day 7, OT-1 CD45.1$^+$ T cells were in vitro activated for 48 h with cis mAb coated microbeads, whereas OT-1 CD45.2$^+$ T cells were activated with trans mAb coated microbeads (microbeads coated with anti-CD3 plus microbeads coated with anti-CD137). On day 9, a mixture of $3.5 \times 10^6$ of OT-1 CD45.1$^+$ and $3.5 \times 10^6$ OT-1 CD45.2$^+$ CD8$^+$ T cells were intravenously injected in 50 μL of saline in tumor-bearing Rag1$^{-/-}$ mice.

**Processing of mouse tissues.** At the indicated time points, liver, spleen, bone marrow and lymph nodes (LNs) were excised. Livers were incubated with 400 MandL/ml of collagenase and 50 μg/ml of DNase (Roche) for 30 minutes at 37 °C. Then, all the specimens were mechanically disaggregated and filtered through a 70 μm cell strainer (Thermo Fisher Scientific). Intrahepatic lymphocytes were isolated after a Percoll (GE Healthcare) gradient centrifugation of liver suspensions. All samples were then stained for flow cytometry. For studies requiring peripheral blood, 100-150 μL of peripheral blood samples were collected on 50 μL of 5% Heparin (Hospira) at the indicated time points.

**IFNγ, IL-2 and Granzyme B measurement by ELISA.** IFNγ, IL-2 and granzyme B concentrations were measured in supernatants from in vitro activated human lymphocytes. BD OptEIA Set Human IFNγ (555142, BD Bioscience), human BD OptEIA Set Human IL-2 (555190, BD Bioscience) and Human Granzyme B ELISA development kit (3485-1H-6, Mabtech) were performed according to the manufacturer's instructions.

**Immunoblotting.** Human CD8$^+$ T cells were stimulated with mAb coated microbeads for 48 hours. Cells were normalized by equal cell number, harvested and lysed in ice-cold radioimmunoprecipitation assay (RIPA) buffer (Sigma-Aldrich), supplemented with protease and phosphatase inhibitors (GE Healthcare), for 30 min on ice. Proteins were separated by SDS-PAGE, and transferred to polyvinylidene difluoride membranes (Millipore). Membranes were blocked in TBS containing 0.05% Tween-20 and 5% non-fat milk for 1 h. Primary antibodies against IKBα (rabbit polyclonal, 1.1000, ab32518, abcam), p65 (mouse monoclonal, 1:5000, Santa Cruz), p52/p100 (rabbit polyclonal, 1.1000, 4882 S, Cell signaling), β-actin (1:1000, A2066, Sigma) and C23 (mouse monoclonal, 1.1000, Santa Cruz) were used. Secondary goat anti-rabbit (1:5000, 170-6515, BIO-RAD) and rabbit antimouse (1:40000, A2304, SIGMA) antibodies were used. Immunoblots were developed with the. Bands were quantified by densitometry using Image Studio Lite v5.2.5 software.

**EpCAM and 5T4 CRISPR/Cas9 gene targeting.** The human gRNAs were designed with the Benchling online software (https://benchling.com). Full-length gRNA sequences were g5T4.1: 5'-CAGGTTGCGGAAGGACACGT-3′ and g5T4.2: 5'-CGTTAACCGCAATCTGACCG-3′; gEpcam.2: 5'-GTGCACCAACTGAAGT ACAC-3′ and gEpcam.3: 5'-GATCCTGACTGCGATGAGAG-3′. Briefly, we designed forward and reverse overlapping oligonucleotides that contain the target DNA sequence (Supplementary Table 4), so the GeneArt Precision gRNA Synthesis Kit (A29377, Invitrogen) can be used to generate a gRNA DNA template containing a T7 promoter by PCR. Subsequent in vitro transcription (IVT) of the gRNA template was followed by spin column purification.

Cas9 ribonucleoproteins (RNPs) were prepared immediately before experiments by incubating TrueCut Cas9 Protein v2 (A36498, Invitrogen) with sgRNA at 1:1 ratio in buffer R at room temperature for 15 min. $10^5$ tumor cells were electroporated with a Neon Transfection System 10 μl kit (MPK1025, Invitrogen). After electroporation, cells were immediately transferred into one well of the 6-well culture plate containing 3000 μL of pre-warmed culture medium without antibiotics.

**Single-cell sorting of CRISPR/Cas9-silenced variants.** For single-cell cloning, single cells were sorted into 96-well plates with 150 μl complete medium. These plates were briefly centrifuged and incubated at 37 °C, 5% CO$_2$, the single-cell clones were evaluated 3 days after sorting to exclude multiple cell contamination. Cells were cultured until confluence. For flow cytometry analysis, cells were harvested with Accutase (Gibco) and stained for EpCAM and 5T4 expression. All samples were acquired on a BD Canto II. All analyses were performed using Flowjo software v.10.6.2 (Tree Star).

**Cleavage assay.** The different tumor samples were cultured and harvested for genomic DNA extraction with the GeneArt Genomic Cleavage Detection Kit (A24372, Invitrogen). Briefly, wild-type and genetically modified tumor cell lines were digested in lysis buffer and genomic DNA samples were amplified by primers flanking the sgRNA targeting site (Supplementary Table 5). The PCR product is denatured and re-annealed so that mismatches are generated as strands with an indel re-annealed to strands with no indel or a different indel. The PCR products were digested with the detection enzyme for 1 h at 37 °C and separated by 2% agarose gel electrophoresis.

**Microarray data acquisition and data processing.** Human CD8$^+$ T cells from three independent donors were stimulated for 32 h with mAb coated beads. RNA was extracted by Trizol (Sigma) purification, whereas the aqueous phase was directly transferred into RNAeasy mini kit columns (74104, Qiagen). The concentration of small quantities of RNA was determined using Nanodrop spectrophotometry. Isolated RNA was sent for RNA quality assessment with a Bioanalyzer and samples with RNA integrity numbers (RIN) > 8 were hybridized to Human Clariom S profiling (Applied Biosystems) microarrays gene expression at the Genomic Department of Murcia University (Spain).

Normalization of microarray data was performed with RMA[49]. After quality assessment using R/Bioconductor v3.5.1[50], a filtering process was carried out to eliminate low expression probe sets. LIMMA (Linear Models for Microarray Data)[51] was used to identify the probe sets that showed significant differential expression between experimental conditions. The consistency of differential gene expression (DGE) was determined by an independent one-sample t test. Only those genes which showed a minimum log2 FC > ± 0.5 and P < 0.01 were considered a DEG.

Ingenuity Pathway Analysis (IPA) software was used to identify canonical signaling pathways, upstream regulators and downstream disease/function pathways associated with the expression profiles of CD137 cis- versus trans costimulation. Fisher's exact test was used to calculate a p value determining the probability that the association between the genes in the dataset and the canonical pathway, upstream regulator or downstream disease/functions could be explained by chance alone. A p value <0.01 was used as the cut-off for identifying significant altered canonical pathways, upstream regulators or downstream disease/functions. In line with IPA cut-off values, z scores of >2·0 or < −2·0 were considered significant activation scores.

**NF-κB luciferase reporter assay.** Luciferase reporter assays were performed with the human Jurkat cell line that has been genetically modified to express hCD137 and a luciferase reporter driven by an NF-κB-responsive element that can respond to CD137 ligand/agonist antibody stimulation (4-1BB Bioassay, JA2351, Promega). Jurkat cells were stimulated for 6 hours with mAb and CD137L-Fc coated beads and lysed in dual reporter lysis buffer (Promega). $12.5 \times 10^4$ wild-type, Trans1 and Trans2 HCT116 cells were seeded in a 96-well plate. After overnight culture, CD137-Jurkat cells were added in the presence of 1, 0.5 and 0.1 μg/ml CD3-EpCAM and 1.5 μg/ml of ALG.APV-527 BsAbs for 6 hours and lysed in dual reporter lysis buffer (Promega). Firefly luciferase signal was detected in an Orion L Microplate Luminometer (Berthold Detection System)

**In vivo NF-κB luciferase reporter assay.** 6–8 weeks old Rag2$^{-/-}$IL-2Rγc$^{-/-}$ were injected subcutaneously with $5 \times 10^5$ wild-type HCT116 cells or with a mixture of 1:1 ratio ($2.5 \times 10^5$: $2.5 \times 10^5$) of HCT116 5T4 KO EpCAM clon3 and HCT116 KO 5T4 clon1. At day 9, $4 \times 10^6$ CD137-Jurkat NF-κB- reporter cell line with 1 μg/tumor of CD3-EpCAM and 10 μg/tumor of ALG.APV.527 BsAbs were intratumorally administered together with the CD137-Jurkat cell line. Bioluminescence imaging (PhotonIMAGER) was performed before (baseline) and 6 hours after adoptive cell transfer and treatment with the bispecific constructs. As control, a group of tumor-bearing mice was treated with CD137-Jurkat NF-κB reporter cell line only and left untreated with the bispecific constructs.

**p65 nuclear translocation assessment by confocal microscopy.** Human CD8$^+$ T cells purified using immunomagnetic separation (MACS) were stimulated with mAb coated microbeads for 48 hours. Cells were then PFA fixed by adding 50 μl of 4% PFA (Thermo). Cells were surface stained with anti-CD8-AF488. Cells were then permeabilized with 0.2% Triton X-100 (Sigma) for 5 min at room temperature. For intranuclear staining anti-p65 (Santa Cruz Technologies) was used, followed by the addition of anti-rabbit AF647 secondary antibody (1964354, Invitrogen) Thermo). Cells were then resuspended in a 2x agar solution containing Hoescht (Life Technologies) and transferred into an 8 well microscopy plate (IBIDI).

Confocal microscopy was performed to quantify total and nuclear p65 signals. Images were taken in an LSM800 equipped with 405, 488, 640 Diodes and a 63x Plan-Apochromat 1, 40 Oil DIC III. Intranuclear staining of p65 was quantified with FIJI by applying ROIs on Hoescht based nuclear shape and on CD8 signal surrounding each CD8$^+$ T cell. Cells that have their nuclear shape compromised were excluded from the analysis.

**Live microscopy of CD8$^+$ T cells interacting with CIS/TRANS beads.** Anti-CD3 mAb (BE0001-2, Bio X cell) and anti-CD137 (6B4, in house) mAb were labeled with Alexa Fluor 647 and Alexa Fluor 488 respectively using Alexa Fluor antibody labeling kits (A20186, A20181, Thermo) prior to conjugation to M450 tosylactivated beads as previously described. CD8$^+$ T cells were preactivated overnight with 1 μg/ml plate-bound αCD3 mAb. Cells were then stained with the fluorescent probe CMRA (Thermo) for 20 min at 37 °C in complete RPMI media at a density of $0.75 \times 10^6$ cells/ml. Fluorescent antibody-coupled beads were then added at 3:1 ratio (beads:T cells).

The microfluidic device for cell confinement was made in Polydimethylsiloxane (PDMS) by replica-molding technique including seven 27 mm long channels with forty-two 250x250x250 μm cubic wells per row and a commercial glass cover-slip,

bonded by plasma oxygen. For the experiments using magnetic beads, 500 μl of Isopropyl Alcohol (IPA) were manually introduced into the device and later removed by flowing RPMI medium from the inlets during two minutes. Then, cells were introduced using a syringe pump at 100 μl/min with a concentration of $1 \times 10^6$/ml until the channels were completely filled up.

For experiments in which Trans and Cis stimuli was provided by tumor cells, HCT116 5T4$^{-/-}$, EpCAM$^{-/-}$ and WT were labeled with CellTracker green CMFDA, CellTracker Deep Red (all from Thermo) or a mixture of both dyes respectively according to manufacturer's instructions. A surface treatment was applied, between four and eight hours before cell loading, in order to promote cell attachment to the inner surfaces of the wells. After washing the channels with IPA, Collagen Type I (Sigma) was manually introduced and left in the device channel for thirty minutes. Collagen was then removed from the device by manually flowing 500 μl of PBS, 500ul of deionized water, and finally 500 μl of RPMI. Then, tumor cells were introduced in the system with a syringe pump at 100 μl/min at a density of $2 \times 10^6$. The cells were left to attach to the well surface for at least four hours. Before loading the lymphocytes mixed with medium, remaining tumor cells in the channels were removed by flowing 1 ml of mounting medium at 200 μl/min. Once the device was ready, CMRA (Thermo) labeled CD8$^+$ T lymphocytes were introduced with a syringe pump at 100μl/min and at a concentration of 0.75 $10^6$ cell/ml.

Live microscopy was performed in a LSM880 inverted confocal microscope (Zeiss) equipped with an incubator to maintain 5%CO$_2$, humidity and 37 °C of temperature throughout the experiment. Images were taken using a 25x water plan apochromat LD objective (NA 0.8) water immersion objective. An Argon 488 laser cell line and two He/Ne lasers (543 and 633 nm) were used to simultaneously excite the three dyes. $512 \times 512$ pixels 3D Z stacks of approximately 20 micrometers were obtained by acquiring images every 1 μm. Two microwells of each CIS and TRANS condition were simultaneously imaged in each experiment.

Time-lapse videos were then analyzed using Imaris software v9.8 (Bitplane). CD8$^+$ T cells (n = 25–50) were semi-automatically tracked using the Spot tool and the tracking algorithms of Imaris software. The number and time of interactions of each CD8$^+$ T cell with different fluorescently labeled beads or cells were quantified based on the movement tracks and confirmed by a second blinded researcher.

**Statistical analysis.** Statistical analyses were performed with GraphPad Prism v8.2.1 using one-way ANOVA and Student's t-tests, as appropriate and indicated in each figure. Significant differences were marked on figures legends as *<0.05, **<0.01 and ***<0.001.

**Reporting Summary.** Further information on research design is available in the Nature Research Reporting Summary linked to this article.

## Data availivity

Microarray data have been deposited in the National Center for Biotechnology Information Gene Expression Omnibus (NCBI-GEO) under primary accession number GSE158041. The remaining data supporting the findings of this study are available within the Article, Supplementary Information or in the Source Data file. Source data are provided with this paper.

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

## Acknowledgements

This work was supported by Alligator Bioscience and Aptevo Therapeutics, Spanish Ministry of Economy and Competitiveness (MINECO SAF2014-52361-R and SAF 2017-83267-C2-1R and PID2020-112892RB-100 [AEI/FEDER,UE]), Cancer Research Institute under the CRI-CLIP, Asociación Española Contra el Cancer (AECC) Foundation under Grant GCB15152947MELE, Joint Translational Call for Proposals 2015 (JTC 2015) TRANSCAN-2 (code: TRS-2016-00000371), projects PI14/01686, PI13/00207, PI16/00668, PI19/01128, funded by Instituto de Salud Carlos III and co-funded by European Union (ERDF, "A way to make Europe"), European Commission within the Horizon 2020 Programme (PROCROP - 635122), Gobierno de Navarra Proyecto LINTERNA Ref.: 0011–1411, Mark Foundation and Fundación BBVA. I.O, is supported by the AECC Investigator 2020. M.A. is supported by the AECC Investigator 2019. C.O.S. is funded by the Spanish Ministry of Science of Innovation and Universities (RTI2018-094494-B-C22, MCIU/AEI/FEDER, UE). AT has received financial support through "la Caixa" Banking Foundation (LCF/BQ/LR18/11640014). M.F.S. is supported by a Miguel Servet contract from Instituto de Salud Carlos III, Fondo de Investigación Sanitaria (Spain). Esther Guirado is acknowledged for project managing, Dr. Diego Aligani for excellent flow cytometry assistance, Victor Segura and Elisabet Gurceaga for bioinformatic support, the Department of Molecular Biology from the Murcia University for the microarray analysis and Dr. Paul Miller for English editing. We are very grateful to all patients and control volunteers who participated in this study and to all clinical staff who helped with participant recruitment. Some figures contain elements from Servier Medical Art (https://smart.servier.com/) and Biorender (https://biorender.com).

## Author contributions

I.O., A.A., P.B., and I.M. designed experiments. I.O., F.J.V., A.T., M.A., M.C.O., E.B., D.C., P.J., S.S-G, I.E., M.H.N., and A.A. performed the experiments and processed samples. J.M-E. provided the virus stocks. I.O. performed all statistical analyses. I.O., A.A., J.G.V., and I. C-D. analysed the data. I.O., F.J.V., A.T., C.O-S, P.E., S.F., G.H-H, M.H-H., M.E.R-R, M-F.S., P.B., and I.M. discussed the data. I.O., P.B. and I.M. wrote the manuscript. All authors performed a critical revision of the manuscript for important intellectual content and final approval of the manuscript.

## Competing interests

I.M. reports advisory roles with Roche-Genentech, Bristol-Myers Squibb, CYTOMX, Incyte, MedImmune, Tusk, F-Star, Genmab, Molecular Partners, Alligator, Bioncotech, MSD, Merck Serono, Boehringer Ingelheim, Astra Zeneca, Numab, Catalym, and Bayer, and research funding from Roche, BMS, Alligator, and Bioncotech. P.B. reports advisory roles with Tusk and Moderna, research funding from Sanofi, and Bavarian Nordic and speaker honoraria from BMS, MSD, Novartis and AstraZeneca. J.M-E. is a full-time employee of Bavarian Nordic. P.E. and S.F. are employees at Alligator and G.H.H. is an employee at Aptevo. The rest of the authors declare no competing interests.
