## [Peer Review File · Nature Communications]

CD137 (4-1BB) costimulation of CD8 T cells is more potent when provided in cis than in trans with respect to CD3-TCR stimulationREVIEWER COMMENTS

Reviewer #1 (Remarks to the Author):

The manuscript of Otana et al describes a study on CD137 (4-1BB) T cell costimulation with respect to in-cis and in-trans signaling regarding TCR triggering. Antibody coated microbeads with both a-CD137 and a-CD3 antibodies (in-cis) is superior to microbeads coated with either a-CD137 or a-CD3 antibodies (in-trans) in activating human and mouse T-cells. The stimulated T cells are distinct in NF-kB activation, and at the transcriptomic level. After competitive adoptive transfer the in-cis stimulated T cells dominate over in-trans stimulated T cells.

Major comments

1. In the manuscript, the authors show that CD137 co-stimulation provided by beads or tumor cell lines in cis is more potent than co-stimulation provided in trans in eliciting T cell responses. However, it remains unclear whether the observed differences are reflective of a situation in vivo, or are influenced by the experimental settings. Co-stimulation provided in trans via beads may result in suboptimal T cell activation as a result of e.g. sterical hindrance (more beads are present in the trans setting) and/or differences in affinity between beads. The authors should experimentally address these concerns carefully to ensure the observed effects do not result from experimental artefacts. Currently, it is unclear whether the effects are caused by quantitative differences (more antibody available) or by qualitative differences. Dose titrating effects and circumventing possible sterical hindrance should be evaluated.
2. Similarly, comparing cis with in trans stimulation via monolayers of genetically modified tumor cells is also sensitive to in vitro artifacts and are not reflective of an in vivo situation.
3. Using the experimental conditions, the authors determine that CD137 co-stimulation provided in cis is more potent than co-stimulation provided in trans to elicit T cell activation and cytokine responses. However, the functional relevance of these findings remain largely unclear with respect to protection and (therapeutic) efficacy. To address this, the authors could provide e.g. data on the impact of cis / trans co-stimulation on the clearance of target cells in an in vitro and in an in vivo setting, such as an in vivo tumor model. Here, the concerns raised under point 1 and 2 should be considered.
4. It remains unclear why in-cis works better than in-trans. One explanation could be a better formation of the immunological synapse (Quintana et al. PNAS 2007). Did the authors consider such mechanism. This could be experimentally addressed or at least discussed.

Minor comments

- In figure 1 and 7, the authors show that CD137 co-stimulation in cis results in lower Eomes levels in comparison to co-stimulation in trans. This appears at odds with the limited maintenance of trans co-stimulated T cells in vivo, since Eomes has been classically associated with central memory development and longevity of T cell memory. The authors should comment on that.
- In the results section, the main text does not always refer to the correct panels of Figure 1 (starting in line 98). Please match the figure panels with the corresponding sections of the text.
- The authors use both T cell and T-cell in the text (e.g. lines 102 and 109). Please harmonize throughout the manuscript.
- The legends of the supplementary figures are not included in the manuscript files. Please provide the corresponding figure legends.
- In figure 4, the authors use Western blots to determine differences in I κ B α , p65, and p52 levels following CD137 co-stimulation provided in cis or trans. These differences could be quantified (e.g. by determining intensity).
- In Figure 7, the authors employ tumor cell xenografts to study the effects of cis vs. trans co-stimulation in vivo. Given that the trans co-stimulation condition requires the presence of two tumor cell clones with genetic ablation of EpCAM or 5T4, respectively, the authors should provide evidence that both clones are present at similar frequencies in the established tumors at 14 days after injection.
- Typo in Figure 7 h: inyection

Reviewer #2 (Remarks to the Author):

This paper provides experimental evidence that signaling in T cells requires anti-CD3 and anti-CD137 be provided on the same beads or tumor cells (cis) versus separate beads or tumor cells. This concept for TCR/CD28 costimulation was first discussed by Liu and Janeway, PNAS 89: 3845-9, showing that signal 1 (MHCp signal) and signal 2 (B7 family) on the same cell are 80x more effective than when given on separate cells (would be nice to cite this in the introduction). This paper now extends this observation to the CD137 costimulatory pathway. This makes logical sense because we know that TCR signals are needed to induce CD137 and its expression is transient and other work shows that 4-1BB becomes part of the immune synapse, so intuitively one expects the anti-CD137 to be more efficient when given in the same time/place as TCR signal. This paper is focused on providing experimental evidence for this concept.

One control that I think is missing from the experimental design of figure 1 and 2 is the following: Anti-CD137 is not present on resting T cells, so the anti-CD137 or 4-1BBL-Fc beads could not bind to the T cells until T cells have first been activated by the anti-CD3 beads. When the two are provided on the same bead, this is readily achieved. However, if separate beads are provided, then the anti-CD3 beads would need to bind first-and they may not readily detach again given the multivalent interaction-therefore it is possible that the anti-CD3 beads given in trans sterically limit the accessibility of the anti-CD137 beads to the T cells that have already bound the anti-CD3 beads. Therefore, to rule out this more trivial reason for the cis beads working better, it would be nice if the authors could demonstrate that the same T cell binds to both kinds of beads simultaneously. For example, they could allow the beads to interact with the T cells in cis or in trans, fix the conjugates and then stain for the IgG1 and Ig2a antibodies- to demonstrate whether the two beads can in fact engage the same cell. The data in figure 1 and 2 seem quite all or none- ie the system seems to work in cis but not in trans... making me question whether they even get contact with both sets of beads. If its just a question of cis or no signal, then studying the mechanism and what genes are induced is less about cis/trans and more about activated/not activated, so I think this is an important point to clarify.

Figure 3 takes advantage of 2 bispecifics, one linking the antigen 5T4 and 4-1BB and the other linking the TCR to Epcam, to provide two signals-TCR and Costimulatory signal. Then they knockout Epcam or 5T4, so that one needs two different tumor lines interacting with the T cell to get both signals... the design is elegant. Albeit the effect of cis costimulation on CD25 is quite marginal and the difference between cis and trans not very large (Fig 3c). On the other hand, in figure de-f, there again seems to be an all or none effect-whereas the signals work in cis but are almost absent when given in trans. In figure 3 f-g the effect of cis/trans on mitotracker signaling look very similar, albeit the authors find significance on the cis but not the trans conditions, but not a very convincing signal. Again, it would be nice to see perhaps by using different fluorescent labels on the two cell types, if in fact the T cell is interacting in trans with the two different tumor types, or if only cis interactions can take place, so that its really activated/not activated and not about differences in cis/trans signaling but whether conjugates form at all?

With respect to Figure S2c-f- the authors use HT116 tumor that expresses both Epcam and 5T4 and titrate down the dose of a bispecific with anti-CD3/anti-Epcam with and without addition of a bispecific with anti-CD137 and anti-5T4 targeting the tumor. I am not sure I understand this experiment. The anti-CD3/anti-Epcam should bring activated T cells into the tumor- adding the CD137 bispecific with a tumor targeting antibody should add costimulation- but in Figure S2c- the effect of adding the anti-CD137 seems fairly small and in c-f with the exception of one data point, doesn't seem to be affected by dose of the first bispecific- so I am not sure what the authors are trying to show here. If costimulation becomes more important when you titrate down signal 1- why isn't the difference larger as you titrate down first bispecific on adding the second. Please clarify the point of this figure. Note- I couldn't find any figure legends for the supplemental material- this would help with reading the supplement.

It is of interest that the studies with Jurkat (NFkB reporter) in Figure 4 show less of an all or none-effect with Cis/Trans stimulation- I wonder if this is because these cells have been transfected to express CD137 constitutively, so they can signal in response to anti-CD137 alone- as one does not need the TCR signal to induce CD137, whereas with normal T cells, you need CD3 signal at the

same time or just before the anti-4-1BB signal, due to the lack of expression of CD137 in resting normal T cells and its transient nature. So the Jurkat model with constitutive CD137 expression is somewhat different from the models used in figure 1-3.

The tumor model data are not particularly impressive with Jurkat; would have been nice to set up a design more like the one in figure 3 in a mouse in vivo model, although I do not know if this is feasible.

Reviewer #3 (Remarks to the Author):

Otano et. Al. design a series of experiments to address the relative potency of trans-activation of CD137 versus cis-activation (by which they mean proximal co-activation with the T cell receptor). There are multiple scientifically compelling questions to be addressed here in terms of 1) further elucidating the relatively poorly understood downstream signaling from CD137, 2) determining whether activation of CD137 within the core of the T cell synapse affords activation advantages to T cell versus the more diffuse geography of trans-mediated agonist antibody activation, and 3) determining to what extent the cell cycle versus effector function vs metabolic benefits of CD137 activation can be separated based on the nature of the receptor stimulus. The data presented show that cis-activation of CD137 through co-coated beads with anti-CD3 or using CD3-bispecific antibodies can powerfully activate markers of T cell cytokine, proliferation, metabolism and cytotoxic potential, lending credence to their hypothesis. Unfortunately, this paper suffers throughout from a critical flaw. The trans-activation control group, both human and mouse, is weaker almost to a fault relative to established literature with 4-1BB activation. For example:

In Figure 1 – The authors find no activation of Granzyme B by their trans 4-1BB agonist. Further, they claim no enhanced Interferon gamma production beyond anti-CD3 alone. The capacity of 4-1BB agonist antibodies to strongly upregulate both of these hallmarks of T cell effector function has been well established across numerous publications and diverse antibodies targeting at least 4 different epitopes of the molecule in vitro and in vivo in mice and in patients. Something is wrong here with the agonist function of the antibody, the coating on beads, or the engagement of the receptor when the antibody is bead coated. It makes the comparative conclusions of the authors uninterpretable however.

In Figure 2 – similar story – their trans 4-1BBL-Fc beads are functionally dead. No IFN γ or Bcl-XL over anti-CD3 alone. This data is just not consistent with known function of 4-1BBL, well established over decades.

Same problem in Figure 3 – there is no significant benefit to IFN γ (maybe a trend), IL-2, or Granzyme B or mitochondrial reporter dye with a trans agonist antibody. All of which are things that have been demonstrated repeatedly by others and by this same lab previously. Also, the statistically significant comparisons are only between the CD3-EpCam bispecific with or without the 5T4-CD137. It would be important to know if the differences between the CD3/EpCAM + 5T4/CD137 combos in the CIS vs Trans1 groups were statistically significant.

In a way Figure 4 illustrates this problem even more clearly. 4A, done with Jurkat Reporters, shows significant NF κ B activation by the Trans beads but even more so by the CIS. However, Figure 4b, again shows no NF κ B activation at all above background by the trans 5T4-CD137 bispecific. 4A actually reflects credible data in light of the existing literature. 4B, in contrast to all existing literature and figure 4A, shows a purported 4-1BB agonist in trans as incapable of activating NF κ B at all.

In Figure 5 the gene expression data is well presented but, again, there are doubts due to the underperformance of the trans control. What does trans versus control Ig look like – nothing? Also it is surprising there were no significantly downregulated genes. 5d vs 5f is very much the story of 4a versus 4b. In 5d the trans induces substantial T cell proliferation as expected from a CD137 agonist, albeit more so for the cis. When we go to their bispecific system in 5f though, trans 4-1BB activation is background. In the end, I think we have to recognize that the use of bispecific antibodies, in which the affinity of the CD3 or 4-1BB agonist arms is generally tuned quite low, is just not a good way to answer the cis vs. trans activation question in these kind of in vitro studies.

For “Trans 2” - Is it surprising that T cells don't get activated by a low affinity monovalent 4-1BB agonist arm with high single to low double digit nM affinity – especially in a system with no tethered CD3 activation (most CD3 bispecifics are engineered for zero T cell activation as free molecules – some with CD3 affinities as low as 20-50nM). So this group doesn't really inform on the efficiency of cis vs trans 4-1BB activation efficiency at all. In fact, its not even clear in the bispecific system how much 4-1BB these T cells can express without receiving a CD3 signal from a tethered bispecific first.

For “Trans 1” – its clear that the 5T4-CD137 bispecific just isn't activating CD137 the way it should. For example, if you dropped urelumab or CTX-471 into this culture, I'm not convinced the activation level would be any less (and possibly could be more) than what's observed with the “CIS” cells. This activation data is also out of sync with other published CD137 bispecific data in similar systems such as PRS-423.

Other comments:

The DNA damage aspect is one of the more interesting findings here, it would be interesting to read the author's thoughts on the origin of the enhanced damage – higher ROS related to the mitochondrial changes? Alterations in repair?

The discussion around Tbet vs Eomes vs the ratio feels incomplete as both of these transcription factors can be “good guys” vs “bad guys” depending on the context. T cells with higher Tbet/Eomes ratios are not always better and higher Eomes has been associated with higher anti-tumor effector function in some contexts but a more differentiated, even anergic state in others. Eomes is a strong activator of both IFN γ and Granzyme B which always feels incongruous in a number of these figures where Eomes is clear induced by the “trans” CD137 agonist but Granzyme and IFN do not respond in kind.

Reviewer #1 (Remarks to the Author):

The manuscript of Otana et al describes a study on CD137 (4-1BB) T cell co-stimulation with respect to in-cis and in-trans signaling regarding TCR triggering. Antibody coated microbeads with both a-CD137 and a-CD3 antibodies (in-cis) is superior to microbeads coated with either a-CD137 or a-CD3 antibodies (in-trans) in activating human and mouse T-cells. The stimulated T cells are distinct in NF-kB activation, and at the transcriptomic level. After competitive adoptive transfer the in-cis stimulated T cells dominate over in-trans stimulated T cells. Major comments

In the manuscript, the authors show that CD137 co-stimulation provided by beads or tumor cell lines in cis is more potent than co-stimulation provided in trans in eliciting T cell responses. However, it remains unclear whether the observed differences are reflective of a situation in vivo, or are influenced by the experimental settings. Co-stimulation provided in trans via beads may result in suboptimal T cell activation as a result of e.g. sterical hindrance (more beads are present in the trans setting) and/or differences in affinity between beads. The authors should experimentally address these concerns carefully to ensure the observed effects do not result from experimental artefacts. Currently, it is unclear whether the effects are caused by quantitative differences (more antibody available) or by qualitative differences. Dose titrating effects and circumventing possible sterical hindrance should be evaluated.

RESPONSE 1: We have performed experiments to rule out the possibility of steric hindrance of the beads for T-cell contact. Our approach has been to time-lapse image with confocal microscopy the ability of fluorescence-labelled CD8 T cells to interact with different colored beads. A video (supplementary video 1) is included and shows that T cells have plenty of opportunities to interact with antibody-coated beads in cis and in trans. Representative frames of the video are shown in Supplementary Fig.S3a.

Furthermore, we carried out these experiments at different bead to T lymphocyte ratios that show the superiority of cis over trans in terms of CD25 and Bcl-xL induction, as well as proliferation. Given space constraints, we include the results here as Figure 1 for reviewer's inspection.

Figure 1 for reviewer's inspection. Human primary CD8 T cells from a healthy donor were activated with mAb coated microbeads for 96 hours. Beads were added at different ratios with respect to T cells. Flow cytometry measurement of CD25 (a), Bcl-xL (b) staining on FACS-gated CD8 T cells following cis or trans costimulation. c, percentage of dividing VPD-pre-loaded CD8 T cells and stimulated with different ratios of mAb coated beads for 96 hours. VPD: violet proliferation dye.

Moreover, we have coated beads with anti-CD7 mAb that binds to T cells without inducing noticeable costimulation. Supplementary Fig.S1g-j shows that the CD7 mAb coated beads did not interfere with the CD137-costimulation provided in *cis*, further indicating no steric hindrance.

2. Similarly, comparing *cis* with *in trans* stimulation via monolayers of genetically modified tumor cells is also sensitive to *in vitro* artifacts and are not reflective of an *in vivo* situation.

RESPONSE 2: This point is of great interest. To address it, we have performed time-lapse confocal microscopy of the *cis* and *trans* costimulation conditions pertaining to experiments in Figure 3. Supplementary video 2 provides an example in which T cells have the opportunity to interact with both *cis* and *trans* tumor cells that were differentially pre-stained with fluorescent dyes. Representative frames are shown in Supplementary Fig.S5a, and quantitative data are shown in Supplementary Fig.S5b.

To address the *in vivo* relevance, we procured access to vaccinia virus vectors expressing OVA with or without 4-1BBL from Bavarian Nordic. Intravenous infection with these modified vaccinia Ankara virus vectors (MVA) resulted in expansion of OVA-specific CD8 T cells that could be followed by proper H-2K^b-SIINFEKL multimers. Conditions were set up in which OVA and 4-1BBL were in the same virus or in separate ones. Very interestingly, the virus providing OVA and 4-1BBL in *cis* was superior to the *trans* condition already upon priming, but much more efficiently so following boosts. These results were also sustained in the spleen, liver and bone marrow of these mice. Furthermore, the level of DNA damage was reduced in the *cis* CD137-costimulation condition. These data are shown in Figure 7 of the revised version, and in our opinion, they provide compelling *in vivo* evidence in a relevant setting (vaccination with vaccinia virus).

3. Using the experimental conditions, the authors determine that CD137 co-stimulation provided in *cis* is more potent than co-stimulation provided in *trans* to elicit T cell activation and cytokine responses. However, the functional relevance of these findings remain largely unclear with respect to protection and (therapeutic) efficacy. To address this, the authors could provide e.g. data on the impact of *cis* / *trans* co-stimulation on the clearance of target cells in an *in vitro* and in an *in vivo* setting, such as an *in vivo* tumor model. Here, the concerns raised under point 1 and 2 should be considered.

RESPONSE 3: This comment has been firstly addressed by assessing cytotoxicity in xCELLigence experiments that are shown in Supplementary FigureS7f and demonstrate the superior CTL activity of OT-1 T lymphocytes against B16.OVA cells when pre-stimulated by *cis* beads as compared to *trans* beads.

Furthermore, we have performed tumor-infiltration competition experiments for Figure 6h and i studying infiltration of B16.OVA tumors by OT1 cells pre-activated either with *cis* or *trans* microbeads as shown in Figure 7h of the revised version. Harvesting tumors and tumor-draining lymph nodes clearly showed a quantitative dominance of *cis*

costimulated CD45.1⁺ OT-1 over trans-costimulated CD45.2⁺ OT-1. We believe that these findings provide compelling evidence for the superiority of cis costimulation in a relevant engrafted tumor setting.

4. It remains unclear why in-cis works better than in-trans. One explanation could be a better formation of the immunological synapse (Quintana et al. PNAS 2007). Did the authors consider such mechanism. This could be experimentally addressed or at least discussed.

RESPONSE 4: We thank the reviewer for suggesting this potential mechanism. The formation of an Immunological synapse (IS) between CD8 T cells and antigen-presenting cells leads to mitochondrial distribution towards the plasma membrane (Quintana et al., PNAS 2007). To study if the superiority of CD137-cis vs trans co-stimulation was due to an efficient formation of IS, primary human CD8 T cells were activated with beads coupled to fluorescent conjugated antibodies. We observed a preferential mitochondria translocation to the plasma membrane area after TCR stimulation, independent of α CD137 stimulation (Fig2 for reviewer's inspection). CD8 T cells in contact with α CD137 beads (green) show a different pattern of distribution of mitochondria throughout the cytosol. Even though our group and others (Teijeira A, Cancer Immunol Res. 2018; Menk AV, PNAS 2018) have shown that α CD137 stimulation enhances mitochondrial functions in CD8 T cells, in our experimental setting, the mitochondria relocalization to the IS was only triggered by TCR stimulation.

Following the advice from reviewer 2, we include the possibility of regulation of immune synapses in the introduction and discussion.

Figure 2 for reviewer's inspection. Anti-CD3 Alexa Fluor 647 and anti-CD137 Alexa Fluor 488 conjugated antibodies were used to coat cis and trans microbeads. Representative confocal fluorescence images from Mitotracker Green/CelTrackerOrange CMRA-labelled CD8T cells after stimulation with mAb coated beads. Cis beads are shown in yellow, whereas α CD3 mAb coupled beads are shown in red and α CD137 AF647 mAb-coupled beads are shown in green. Mitochondria appear also as green stained structures inside the blue cells.

In fact, we are already working in crosstalk of receptors at the immune synapse involving CD137 signalsomes but such experimentation is mostly unrelated to the scope this manuscript.

Minor comments

- In figure 1 and 7, the authors show that CD137 co-stimulation in *cis* results in lower Eomes levels in comparison to co-stimulation in *trans*. This appears at odds with the limited maintenance of *trans* co-stimulated T cells *in vivo*, since Eomes has been classically associated with central memory development and longevity of T cell memory.

The authors should comment on that.

RESPONSE 5: We have reported our observations on the effects of *cis* as compared to *trans* stimulation. The very subtle balance of these two transcription factors to control differentiation and exhaustion of T lymphocytes is very complex (Pritchard GH, Nat Rev Immunol. 2019). Indeed, the interplay of T-bet and Eomes seems to be tightly co-regulated and the effect of T-bet is different depending on the coexistence of Eomes (McLane L, Cell Reports, 2021). Accordingly, we briefly comment on this subject in discussion, providing these new citations. Needles to remark that this is a very interesting and conflicting research topic and, in fact, we are actively pursuing transcriptomic studies on the consequences of CD137 costimulation.

- In the results section, the main text does not always refer to the correct panels of Figure 1 (starting in line 98). Please match the figure panels with the corresponding sections of the text.

RESPONSE 6: We have corrected these mistakes.

- The authors use both T cell and T-cell in the text (e.g. lines 102 and 109). Please harmonize throughout the manuscript.

RESPONSE 7: We have corrected the text so "T-cell" is only used when acts as an adjective in the sentence, while "T cell" and "T cells" are used as nouns (not hyphenated).

- The legends of the supplementary figures are not included in the manuscript files. Please provide the corresponding figure legends.

RESPONSE 8: We do apologize for this omission, and the revised legends to supplementary figures are enclosed in our revised submission.

- In figure 4, the authors use Western blots to determine differences in I κ B α , p65, and p52 levels following CD137 co-stimulation provided in *cis* or *trans*. These differences could be quantified (e.g. by determining intensity).

RESPONSE 9: These densitometry data are provided in Figure 4a, f and g.

- In Figure 7, the authors employ tumor cell xenografts to study the effects of *cis* vs. *trans* co-stimulation *in vivo*.

Given that the trans co-stimulation condition requires the presence of two tumor cell clones with genetic ablation of EpCAM or 5T4, respectively, the authors should provide evidence that both clones are present at similar frequencies in the established tumors at 14 days after injection.

RESPONSE 10: These results have been moved to Figure 4h, i and j to rearrange our display items. The relative presence of the tumor cell clones in the xenografted tumors is provided in an inset included in the Figure 4h panel.

- Typo in Figure 7 h: inyjection

RESPONSE 11: This typo has been corrected. Thank you.

Reviewer #2 (Remarks to the Author):

This paper provides experimental evidence that signaling in T cells requires anti-CD3 and anti-CD137 be provided on the same beads or tumor cells (cis) versus separate beads or tumor cells. This concept for TCR/CD28 co-stimulation was first discussed by Liu and Janeway, PNAS 89: 3845-9, showing that signal 1 (MHCp signal) and signal 2 (B7 family) on the same cell are 80x more effective than when given on separate cells (would be nice to cite this in the introduction). This paper now extends this observation to the CD137 costimulatory pathway. This makes logical sense because we know that TCR signals are needed to induce CD137 and its expression is transient and other work shows that 4-1BB becomes part of the immune synapse, so intuitively one expects the anti-CD137 to be more efficient when given in the same time/place as TCR signal. This paper is focused on providing experimental evidence for this concept.

One control that I think is missing from the experimental design of figure 1 and 2 is the following: Anti-CD137 is not present on resting T cells, so the anti-CD137 or 4-1BBL-Fc beads could not bind to the T cells until T cells have first been activated by the anti-CD3 beads. When the two are provided on the same bead, this is readily achieved. However, if separate beads are provided, then the anti-CD3 beads would need to bind first-and they may not readily detach again given the multivalent interaction-therefore it is possible that the anti-CD3 beads given in trans sterically limit the accessibility of the anti-CD137 beads to the T cells that have already bound the anti-CD3 beads. Therefore, to rule out this more trivial reason for the cis beads working better, it would be nice if the authors could demonstrate that the same T cell binds to both kinds of beads simultaneously. For example, they could allow the beads to interact with the T cells in cis or in trans, fix the conjugates and then stain for the IgG1 and Ig2a antibodies- to demonstrate whether the two beads can in fact engage the same cell.

RESPONSE 12: This is a key point also raised by reviewer 1. To address this issue, we have performed experiments under live confocal microscopy that show no steric hindrance in the interaction of T cells with both kind of microbeads that were differently labelled with fluorescence. The new data are in Supplementary FigureS3 and Supplementary video 1.

The data in figure 1 and 2 seem quite all or none- ie the system seems to work in cis but not in trans... making me question whether they even get contact with both sets of beads.If its just a question of cis or no signal, then studying the mechanism and what genes are induced is less about cis/trans and more about activated/not activated, so I think this is an important point to clarify.

RESPONSE 14: We chose the T cell:bead ratio of 3:1 because it was giving the largest differences of cis versus trans costimulation. We chose these conditions in experiments such as the one shown below for reviewers' inspection in which the difference was substantiated in all the ratios, but there was also activity in the trans costimulation settings.

Figure 1 for reviewer's inspection. Human primary CD8 T cells from a healthy donor were activated with mAb coated microbeads for 96 hours. Beads were added at different ratios with respect to T cells. Flow cytometry measurement of CD25 (a), Bcl-xL (b) staining on FACS-gated CD8 T cells following cis or trans costimulation. c, percentage of dividing VPD-pre-loaded CD8 T cells and stimulated with different ratios of mAb coated beads for 96 hours. VPD: violet proliferation dye.

Figure 3 takes advantage of 2 bispecifics, one linking the antigen 5T4 and 4-1BB and the other linking the TCR to Epcam, to provide two signals-TCR and Costimulatory signal. Then they knockout Epcam or 5T4, so that one needs two different tumor lines interacting with the T cell to get both signals... the design is elegant. Albeit the effect of cis co-stimulation on CD25 is quite marginal and the difference between cis and trans not very large (Fig 3c). On the other hand, in figure de-f, there again seems to be an all or none effect-whereas the signals work in cis but are almost absent when given in trans. In figure 3 f-g the effect of cis/trans on mitotracker signaling look very similar, albeit the authors find significance on the cis but not the trans conditions, but not a very convincing signal. Again, it would be nice to see perhaps by using different fluorescent labels on the two cell types, if in fact the T cell is interacting in trans with the two different tumor types, or if only cis interactions can take place, so that its really activated/not activated and not about differences in cis/trans signaling but whether conjugates form at all?

RESPONSE 15: First, we examined under live confocal microscopy the interaction of T cells to show that access to cis or trans costimulatory tumor cells seems not to be a limiting factor (Supplementary video 2 and Supplementary FigureS5 a and b).

Having observed so, we provide in Figure 4c several concentrations of CD3-EpCAM stimulating the CD137+ Jurkat reporter cells. Indeed, there is a baseline signal but readily augmented by CD137 co-stimulation both in cis or in trans, but more prominently so when given in cis by ALG.APV-527 (Figure 4c of the revised version).

With respect to Figure S2c-f- the authors use HT116 tumor that expresses both Epcam and 5T4 and titrate down

the dose of a bispecific with anti-CD3/anti-Epcam with and without addition of a bispecific with anti-CD137 and anti-5T4 targeting the tumor. I am not sure I understand this experiment. The anti-CD3/anti-Epcam should bring activated T cells into the tumor- adding the CD137 bispecific with a tumor targeting antibody should add co-stimulation- but in Figure S2c- the effect of adding the anti-CD137 seems fairly small and in c-f with the exception of one data point, doesn't seem to be affected by dose of the first bispecific- so I am not sure what the authors are trying to show here. If co-stimulation becomes more important when you titrate down signal 1- why isn't the difference larger as you titrate down first bispecific on adding the second. Please clarify the point of this figure. Note- I couldn't find any figure legends for the supplemental material- this would help with reading the supplement.

RESPONSE 16: We deeply apologize for not having uploaded legends for supplementary figures. We agree with the reviewer that former Supplementary Figure S2 was not informative and we have removed it from the article.

It is of interest that the studies with Jurkat (NFkB reporter) in Figure 4 show less of an all or none-effect with Cis/Trans stimulation- I wonder if this is because these cells have been transfected to express CD137 constitutively, so they can signal in response to anti-CD137 alone- as one does not need the TCR signal to induce CD137, whereas with normal T cells, you need CD3 signal at the same time or just before the anti-4-1BB signal, due to the lack of expression of CD137 in resting normal T cells and its transient nature. So the Jurkat model with constitutive CD137 expression is somewhat different from the models used in figure 1-3.

RESPONSE 17: The reviewer is right in pointing out this difference that actually supports the concept of cis superiority. Indeed, CD137 stimulation alone induces NF-κB dependent transcription, but TCR-CD3 stimulation with CD137 costimulation attains this effect much more intensely when CD137 costimulation is provided in cis. These data are shown below for reviewer's inspection due to space constraints.

Figure 3 for reviewer's inspection. CD137-Jurkat NF-κB reporter cell line was incubated with anti-CD137 antibody coated microbeads. Luciferase activity was measured after 6 hours.

The tumor model data are not particularly impressive with Jurkat; would have been nice to set up a design more like the one in figure 3 in a mouse in vivo model, although I do not know if this is feasible.

RESPONSE 18: The point raised by the reviewers is addressed in two ways. First, we have shown that OT-1 T lymphocytes that had been co-stimulated in cis, infiltrate B16.OVA tumors and tumor-draining lymph nodes more

abundantly than those which had been co-stimulated in trans (Figure 7h and I of the revised version). Moreover, OT-1 T lymphocytes that had been preactivated in cis killed more efficiently B16.OVA tumor cells in xCELLigence experiments (Supplementary FigureS6f of the revised version).

Reviewer #3 (Remarks to the Author):

Otano et. Al. design a series of experiments to address the relative potency of trans-activation of CD137 versus cis-activation (by which they mean proximal co-activation with the T cell receptor). There are multiple scientifically compelling questions to be addressed here in terms of 1) further elucidating the relatively poorly understood downstream signaling from CD137, 2) determining whether activation of CD137 within the core of the T cell synapse affords activation advantages to T cell versus the more diffuse geography of trans-mediated agonist antibody activation, and 3) determining to what extent the cell cycle versus effector function vs metabolic benefits of CD137 activation can be separated based on the nature of the receptor stimulus. The data presented show that cis-activation of CD137 through co-coated beads with anti-CD3 or using CD3-bispecific antibodies can powerfully activate markers of T cell cytokine, proliferation, metabolism and cytotoxic potential, lending credence to their hypothesis. Unfortunately, this paper suffers throughout from a critical flaw. The trans-activation control group, both human and mouse, is weaker almost to a fault relative to established literature with 4-1BB activation. For example: In Figure 1 – The authors find no activation of Granzyme B by their trans 4-1BB agonist. Further, they claim no enhanced Interferon gamma production beyond anti-CD3 alone. The capacity of 4-1BB agonist antibodies to strongly upregulate both of these hallmarks of T cell effector function has been well established across numerous publications and diverse antibodies targeting at least 4 different epitopes of the molecule in vitro and in vivo in mice and in patients. Something is wrong here with the agonist function of the antibody, the coating on beads, or the engagement of the receptor when the antibody is bead coated. It makes the comparative conclusions of the authors uninterpretable however.

RESPONSE 19: The reviewer makes an important point. We agree that CD137 costimulation also works in trans. We have chosen a CD3 concentration to coat the beads at the conditions in which cis and trans differentiated the most. As we show now in Supplementary FigureS1 a-e, there is costimulation in trans for induction of CD25, Bcl-xl, Ki67, T-bet and Eomes reduction. In all conditions cis was superior, but we chose 2µg for experiments because of being the most clearcut at differentiating cis and trans CD137-costimulation.

Moreover, the ratio of beads to T cells also mattered, and we chose 3:1 ratio according to experiments as the one shown below for reviewer's inspection, which also indicates the superiority of cis versus trans.

In Figure 2 – similar story – their trans 4-1BBL-Fc beads are functionally dead. No IFN γ or Bcl-XL over anti-CD3 alone. This data is just not consistent with known function of 4-1BBL, well established over decades.

RESPONSE 20: Figure 1 and 2 show data as Δ (increment) of expression. We found this useful to focus on the cis versus trans differences, but baseline CD137 costimulation in trans was readily observed. We found this way of presenting the results very useful to normalize variability among human donors.

Same problem in Figure 3 – there is no significant benefit to IFN γ (maybe a trend), IL-2, or Granzyme B or mitochondrial reporter dye with a trans agonist antibody. All of which are things that have been demonstrated repeatedly by others and by this same lab previously. Also, the statistically significant comparisons are only between the CD3-EpCam bispecific with or without the 5T4-CD137. It would be important to know if the differences between the CD3/EpCAM + 5T4/CD137 combos in the CIS vs Trans1 groups were statistically significant.

RESPONSE 21: Indeed, in the tumor cell system redirected signal one with EpCAM-CD3 and 5T4-CD137 only seems to work in cis with differences that were statistically significant and are shown in the figure. There are obvious differences due to heterogeneous human donor variability, but the differential effects of cis costimulation were clear. The requested statistical comparisons are now provided in the Figure3 of the revised version.

In a way Figure 4 illustrates this problem even more clearly. 4A, done with Jurkat Reporters, shows significant NF κ B activation by the Trans beads but even more so by the CIS. However, Figure 4b, again shows no NF κ B activation at all above background by the trans 5T4-CD137 bispecific. 4A actually reflects credible data in light of the existing literature. 4B, in contrast to all existing literature and figure 4A, shows a purported 4-1BB agonist in trans as incapable of activating NF κ B at all.

RESPONSE 22: The reviewer raises an important point. Indeed, trans costimulation of the Jurkat-NF- κ B reporter cells occurred at higher concentrations of EpCAM-CD3, providing signal one. These important data are included in the Figure 4c of the revised version.

In Figure 5 the gene expression data is well presented but, again, there are doubts due to the underperformance of the trans control. What does trans versus control Ig look like – nothing? Also it is surprising there were no significantly downregulated genes. 5d vs 5f is very much the story of 4a versus 4b. In 5d the trans induces substantial T cell proliferation as expected from a CD137 agonist, albeit more so for the cis. When we go to their bispecific system in 5f though, trans 4-1BB activation is background. In the end, I think we have to recognize that the use of bispecific antibodies, in which the affinity of the CD3 or 4-1BB agonist arms is generally tuned quite low, is just not a good way to answer the cis vs. trans activation question in these kind of in vitro studies.

RESPONSE 23: Our previous responses should be convincing in the sense that trans costimulation clearly was detectable even if always weaker as compared to cis costimulation. In the transcriptomic analyses we did not focus on downregulated genes since we did not find any with a fold change greater than $-\text{Log}(0.5)$.

CD137 costimulation in trans provided transcriptomic changes that we are currently studying and will be part of a separate manuscript, since some of the transcriptional targets are known, but some are not.

For "Trans 2" - Is it surprising that T cells don't get activated by a low affinity monovalent 4-1BB agonist arm with high single to low double digit nM affinity – especially in a system with no tethered CD3 activation (most CD3 bispecifics are engineered for zero T cell activation as free molecules – some with CD3 affinities as low as 20-50nM). So this group doesn't really inform on the efficiency of cis vs trans 4-1BB activation efficiency at all. In fact, its not even clear in the bispecific system how much 4-1BB these T cells can express without receiving a CD3 signal from a tethered bispecific first.

RESPONSE 24: This EpCAM-CD3 is not a clinical agent and has good affinity but requires binding to solid phase to crosslink CD3. In Figure 4c, we show on Jurkat reporter cells different concentrations of CD3-EpCAM that clearly show the costimulation in trans 1 and trans 2 conditions. We thank the reviewer for raising this point.

For "Trans 1" – its clear that the 5T4-CD137 bispecific just isn't activating CD137 the way it should. For example, if you dropped urelumab or CTX-471 into this culture, I'm not convinced the activation level would be any less (and possibly could be more) than what's observed with the "CIS" cells. This activation data is also out of sync with other published CD137 bispecific data in similar systems such as PRS-423 (anti Her2-anti CD137).

RESPONSE 25: The CD137 agonists in the clinic or entering the clinic work in cis or in trans. Effects in trans are expected and have been observed by us with a FAP targeted 4-1BBL in humans. In fact, sufficient density of CD137

ligation could perhaps compensate for the costimulation provision in trans. This is commented on the revised discussion.

In our new experiments with MVA vaccines providing CD137L in cis or in trans with respect to the OVA antigen, we also found superiority in cis following priming and more strongly so following boost (Figure 7 of the revised version).

Other comments:

The DNA damage aspect is one of the more interesting findings here, it would be interesting to read the author's thoughts on the origin of the enhanced damage – higher ROS related to the mitochondrial changes? Alterations in repair?

RESPONSE 26: We agree that preventing DNA damage is important, and we are focusing our research on these aspects. Indeed, we identified a number of genes that function in DNA repair pathways. We surmise that DNA damage in T cells undergoing activation and clonal expansion mainly comes from mitochondria ROS as we have recently reported in human T lymphocytes (Otano I, Theranostics, 2021). We touch briefly the point in the discussion of the revised manuscript.

Rapid DNA active replication also causes mistakes that, if not properly and swiftly repaired, cause trouble for the proliferating T-cell clone in physiological conditions (McNally JP, PNAS, 2017). As mentioned, this is the subject of a complex research project currently ongoing in our laboratory.

Regarding the DNA damage reduction effect in cis, we provide now in vivo data upon immunization with ova-encoding vaccinia viruses that provide CD137L costimulation either in cis or in trans. It is reassuring to report that in this in-vivo situation cis costimulation resulted in less DNA damage signal in OVA-specific CD8 T lymphocytes (Figure 7 panel d of the revised version).

The discussion around Tbet vs Eomes vs the ratio feels incomplete as both of these transcription factors can be "good guys" vs "bad guys" depending on the context. T cells with higher Tbet/Eomes ratios are not always better and higher Eomes has been associated with higher anti-tumor effector function in some contexts but a more differentiated, even anergic state in others. Eomes is a strong activator of both IFN γ and Granzyme B which always feels incongruous in a number of these figures where Eomes is clearly induced by the "trans" CD137 agonist but Granzyme and IFN do not respond in kind.

RESPONSE 27: We cannot agree more with the comment raised by the reviewer. We understand the Tbet/Eomes balance as the key point in regulation, but the complexity of this system goes far beyond the scope of our paper. We only report the differences in cis versus trans without entering into the subtle transcriptional consequences of such changes. We provide references (McLane L, Cell Reports 2021; Pritchard GH, Nat Rev Immunol. 2019) for discussion in the revised version of the manuscript.

REVIEWER COMMENTS

Reviewer #1 (Remarks to the Author):

The authors have addressed the comments. I suggest to include the provided Figure 1 (displayed in the rebuttal letter) into the manuscript as it is informative (at least as supplementary Figure).

Reviewer #2 (Remarks to the Author):

The authors have made substantial attempt to address reviewer concerns.

The addition of the in vivo viral vector model, clearly supports that cis delivery of Ag and 4-1BBL is more efficient than trans delivery in two different vectors, albeit its such a different model, it does not really relate to the model used to study the signaling induced by cis vs. trans delivery in vitro. It would also be nice to use this model with an OVA containing tumor to show that it results in improved tumor control in vivo. Also, reporting T cell data only based on frequency can sometimes be misleading if denominator changes-could the authors also report total numbers of Ag-specific CD8 T cells recovered in each experimental group.

The in vitro studies of signaling are largely based on the bead model. The authors have added substantial data to rule out steric hindrance as a trivial explanation of the results. One thing I did note however, was that the CD8 T cells seem to interact only very transiently with the anti-CD3 beads in the video but longer with the anti-CD137 beads and even longer with the cis beads. Does this reflect biology or the affinity/avidity of the antibody used. This was less obvious in the tumor video, but I also had the impression interactions with anti-CD3+ tumors were more transient than with anti-CD137 tumors or cis tumors. It would be helpful if the authors could quantify time of the interactions in the video and report that as well as the number of interactions, and perhaps comment on the avidity issue. On line 185 the statement is made that these experiments rule out less frequent interactions as a trivial answer-but the time of the interaction seems to be different. We know from studies of weak agonist peptides that short duration TCR interactions can lead to incomplete activation compared to stable interactions. So was not sure if it's the time required for productive interaction that is too low for anti-CD3 alone and is this due to the low avidity antibody, but adding anti-CD137 in cis, allows for longer/repeat interactions? so perhaps the signaling differences they measure just reflect poor TCR signaling in the model.

Reviewer #3 (Remarks to the Author):

I appreciate the additional work of the authors but I believe this system is still flawed as pertains to activation by the trans 4-1BB. The authors show bead contact can be simultaneous, which is helpful, but still - a substantial number of T cells still get a mono-CD3 signal absent co-stimulation which may render them insensitive / less fit to respond to future encounters with the 4-1BB beads or both. This does not reflect the in vivo situation where T cells will have been activated with both signal 1 and signal 2 prior to receiving a 4-1BBL - 4-1BB signal.

I also agree with Reviewer 1 that the data on Eomes appears inconsistent as well. Among other things, the delta Eomes MFI changes in 1K may be statistically significant but they are likely physiologically meaningless at those levels.

The conclusion that cis co-stimulation is more potent than trans may still be valid given the data so I leave the decision on publication to the other reviewers and editors.

POINT BY POINT REPLY:

Reviewer #1 (Remarks to the Author):

The authors have addressed the comments.

1. I suggest to include the provided Figure 1 (displayed in the rebuttal letter) into the manuscript as it is informative (at least as supplementary Figure)

RESPONSE 1: we appreciate the reviewer's suggestion and, accordingly, we have included those new data as panels f to j in Supplementary Figure1.

Reviewer #2 (Remarks to the Author):

The authors have made substantial attempt to address reviewer concerns.

2. The addition of the in vivo viral vector model, clearly supports that cis delivery of Ag and 4-1BBL is more efficient than trans delivery in two different vectors, albeit its such a different model, it does not really relate to the model used to study the signaling induced by cis vs. trans delivery in vitro. It would also be nice to use this model with an OVA containing tumor to show that it results in improved tumor control in vivo.

RESPONSE 2: We evaluated the cis versus trans CD137L-immunization with the vaccinia virus vectors in a established tumor mice model. Administration of OVA encoding Modified Vaccinia Ankara vectors that encode CD137L (MVA-OVA-4-1BBL) either in the same vector (cis) or when it is provided by separate vectors (MVA-OVA + MVA-4-1BBL) (trans) resulted in potent antitumor effects against B16.OVA melanomas (Figure 1 for reviewer's inspection). We observed a slightly but statistically significant longer delay in the tumor growth of mice immunized with MVA-OVA-4-1BBL as compared with the group of mice immunized with combined MVA-OVA + MVA-4-1BBL vectors. However, we believe that the intravenous administration of the different MVA constructs is not optimal for a therapeutic evaluation of the cis versus trans CD137L-immunization differences, since it has been shown that the intratumoral administration of MVA constructs exerted strong therapeutic responses in various unrelated tumor models (Medina-Echeverz, J, Nat Comm, 2019).

[Redacted]

3. , reporting T cell data only based on frequency can sometimes be misleading if denominator changes-could the authors also report total numbers of Ag-specific CD8 T cells recovered in each experimental group.

RESPONSE 3: According to the reviewers' suggestion, absolute numbers of OVA-specific CD8 T cells were increased in the blood after MVA vaccination with CD137L costimulation in cis more abundantly than the combination of viruses providing CD137L costimulation in trans (Figure 2 for reviewer's inspection). Absolute numbers of OVA-specific CD8 T cells could not be accurately determined from the cell suspensions prepared from other organs due to dissimilarity numbers of events acquired in the flow

cytometry determinations. In any case, the reviewer is right and differences were also observed in terms of absolute numbers.

Figure 2 for reviewer's inspection. C57/Bl6 mice were immunized with 5×10^7 TCID₅₀ of the different rMVAs at day 0 and day 14 (n=7 for PBS, n=6 for rMVA-OVA, n=7 for cis-costimulation and n=7 for trans-costimulation). **a**, absolute numbers of OVA-specific CD8 T-cell response were quantified by tetramer staining of 100ml of peripheral blood CD8 T cells at the indicated timepoints.

4. The in vitro studies of signaling are largely based on the bead model. The authors have added substantial data to rule out steric hindrance as a trivial explanation of the results. One thing I did note however, was that the CD8 T cells seem to interact only very transiently with the anti-CD3 beads in the video but longer with the anti-CD137 beads and even longer with the cis beads. Does this reflect biology or the affinity/avidity of the antibody used. This was less obvious in the tumor video, but I also had the impression interactions with anti-CD3+ tumors were more transient than with anti-CD137 tumors or cis tumors. It would be helpful if the authors could quantify time of the interactions in the video and report that as well as the number of interactions, and perhaps comment on the avidity issue. On line 185 the statement is made that these experiments rule out less frequent interactions as a trivial answer-but the time of the interaction seems to be different. We know from studies of weak agonist peptides that short duration TCR interactions can lead to incomplete activation compared to stable interactions. So was not sure if it's the time required for productive interaction that is too low for anti-CD3 alone and is this due to the low avidity antibody, but adding anti-CD137 in cis, allows for longer/repeat interactions? so perhaps the signaling differences they measure just reflect poor TCR signaling in the model.

RESPONSE 4: We are grateful to the reviewer because of raising this relevant point. To address it, we have quantified the contact duration of beads:T cells and tumor cells:T cells upon cis and trans costimulation in the time-lapse confocal microscopy videos. The time interval of T-cell interaction with cis-colored beads is similar to the time of interaction with the trans-colored beads. Moreover, we also found that the duration of the T-cell contact with labeled tumor cells is similar under both cis and trans costimulation conditions, thereby ruling out potential artifacts.

These quantitative data are included in the revised Supplementary Figure 4c and Supplementary Figure 6c. The methodology is now described in Materials and methods.

Reviewer #3 (Remarks to the Author):

5. I appreciate the additional work of the authors but I believe this system is still flawed as pertains to activation by the trans 4-1BB. The authors show bead contact can be simultaneous, which is helpful, but still - a substantial number of T cells still get a mono-CD3 signal absent co-stimulation which may render them insensitive / less fit to respond to future encounters with the 4-1BB beads or both. This does not reflect the in vivo situation where T cells will have been activated with both signal 1 and signal 2 prior to receiving a 4-1BBL - 4-1BB signal.

RESPONSE 5: This point is of great interest for us. Indeed, priming in physiological conditions conceivably involves CD28 costimulation. In order to address the point raised by the reviewer, we have pre-activated CD8 T cells with plate-bound anti-CD3 mAb and soluble anti-CD28 mAb for 16h. Such CD8 T cell cultures were left to rest for 24 hours without stimulation, and then re-stimulated with the cis and trans antibody-coated microbeads for 96 hours. The new Supplementary Figure 3 of the revised clearly shows again the superiority of CD137-cis versus trans costimulation on these CD8 T lymphocytes primed under CD28 costimulation in terms of activation markers and interferon- γ secretion. We are grateful for this suggestion and the new experiments are described in the results section of the revised version.

6. I also agree with Reviewer 1 that the data on Eomes appears inconsistent as well. Among other things, the delta Eomes MFI changes in 1K may be statistically significant but they are likely physiologically meaningless at those levels.

RESPONSE 6: We found a slight downregulation on the gMFI of Eomes in CD8 T cells stimulated with cis beads as compared to trans stimulation. It has been proposed that altering the strength of TCR signalling may increase the proportion of T-bet relative to Eomes (Nayar R, *PLOS ONE*, 2015). These results suggest that the TCR and costimulation signals play an important role in regulating Eomes expression. However, we agree with the reviewer that these results could mean a weak physiological effect. In any case, the focus of our manuscript is not to study this complex EOMES/T-BET transcriptional system as we already stated in the discussion section taking into account the previous points raised by the reviewer.

The conclusion that cis co-stimulation is more potent than trans may still be valid given the data so I leave the decision on publication to the other reviewers and editors.

RESPONSE: We indeed agree with the reviewer on the validity of the conclusions in the sense that cis 4-1BB costimulation provides different quantitative and qualitative outcomes, that we consider are demonstrated to be superior to those observed following trans costimulation. We have done our best to provide human and mouse conclusive results in vitro and reveal some of the consequences in vivo. The translational relevance of our work for 4-1BB-based cancer immunotherapies and the gain knowledge justify publication in our opinion.

REVIEWERS' COMMENTS

Reviewer #2 (Remarks to the Author):

The authors have adequately addressed all remaining concerns.
I believe the work is suitable for publication in Nature communications.